# CoSpaDi: Compressing LLMs via Calibration-Guided Sparse Dictionary Learning

## Abstract

Post-training compression of large language models (LLMs) largely relies on low-rank weight approximation, which represents each column of a weight matrix in a shared low-dimensional subspace. While this is a computationally efficient strategy, the imposed structural constraint is rigid and can lead to a noticeable model accuracy drop. In this work, we propose **CoSpaDi** (**Co**mpression via **Spa**rse **Di**ctionary Learning), a novel training-free compression framework that replaces low-rank decomposition with a more flexible structured sparse factorization in which each weight matrix is represented with a dense dictionary and a column-sparse coefficient matrix. This formulation enables a union-of-subspaces representation: different columns of the original weight matrix are approximated in distinct subspaces spanned by adaptively selected dictionary atoms, offering greater expressiveness than a single invariant basis. Crucially, **CoSpaDi** leverages a small calibration dataset to optimize the factorization such that the output activations of compressed projection layers closely match those of the original ones, thereby minimizing functional reconstruction error rather than mere weight approximation. This data-aware strategy preserves better model fidelity without any fine-tuning under reasonable compression ratios. Moreover, the resulting structured sparsity allows efficient sparse–dense matrix multiplication and is compatible with post-training quantization for further memory and latency gains. We evaluate **CoSpaDi** across multiple Llama and Qwen models under per-layer and per-group settings at $20 - 50\%$ compression ratios, demonstrating consistent superiority over state-of-the-art data-aware low-rank methods both in accuracy and perplexity. Our results establish structured sparse dictionary learning as a powerful alternative to conventional low-rank approaches for efficient LLM deployment.

## 1 Introduction

Large Language Models (LLMs) have demonstrated remarkable performance across a wide range of applications, from conversational agents (Brown et al., 2020; OpenAI, 2023) to general-purpose reasoning systems (Touvron et al., 2023a; Anil et al., 2023), owing to their ability to capture long-range dependencies through attention mechanisms (Vaswani et al., 2017; Devlin et al., 2019). However, this success entails substantial memory and computational demands during both training and inference, which severely limits their deployment in resource-constrained environments.

Various compression and acceleration techniques have been proposed over recent years to mitigate these challenges, including pruning (Frankle & Carbin, 2019; Sanh et al., 2020), quantization (Dettmers et al., 2022; Xiao et al., 2023; Yao et al., 2022; Dettmers et al., 2023), and weight decomposition/factorization (Denton et al., 2014; Hu et al., 2022; Liu et al., 2024; Zhang et al., 2023; Ma et al., 2019; Chen et al., 2018; Hsu et al., 2022; Yu & Wu, 2023; Wang et al., 2025a).

Recently, there has been growing interest in effective compression of large models without retraining or fine-tuning, since this is often computationally prohibitive. A prominent line of work leverages low-rank approximations of weight matrices via Singular Value Decomposition (SVD). Early approaches such as DRONE (Chen et al., 2021), FWSVD (Hsu et al., 2022) and ASVD (Yuan et al., 2023) introduced activation-aware truncation of singular values, while SVD-LLM (Wang et al., 2025a) and its variant SVD-LLMv2 (Wang et al., 2025b) proposed truncation-aware and optimized singular value selection strategies. In parallel, Basis Sharing (Wang et al., 2024) explored cross-

layer sharing of a common basis to further enhance compression efficiency by exploiting inter-layer redundancy. Despite their effectiveness, these methods are inherently constrained by the use of a single shared low-dimensional subspace, which may limit representational flexibility and prevent full exploitation of redundancies in LLM parameters.

In this work, we argue that moving beyond the low-rank weight decomposition is a promising new direction. Specifically, we propose to adopt dictionary learning, a well-established paradigm in signal and image processing (Aharon et al., 2006; Elad, 2010), and apply it to LLM compression. Unlike low-rank approximations that confine all columns of a weight matrix to a shared linear subspace, dictionary learning enables a richer union-of-subspaces representation: each column is approximated using only a sparse subset of atoms from a learned dictionary, allowing different columns to reside in distinct subspaces spanned by different atom combinations. Such a representation better accommodates heterogeneous features and introduces additional flexibility to reduce the overall approximation error.

To this end, we introduce **CoSpaDi** (**Co**mpression via **Spa**rse **Di**ctionary Learning), a training-free framework that applies dictionary learning to jointly estimate dictionaries and sparse coefficients used for compressing LLM weight matrices. Our contributions are as follows:

- We introduce dictionary learning with sparse coding as a new paradigm for LLM compression, addressing the limitation of using a single invariant basis for the weight approximation which is inherent in SVD-based methods.

- We demonstrate that **CoSpaDi** is effectively integrated with data-aware optimization, yielding state-of-the-art compression performance for a wide range of compression ratios, while being compatible with post-training quantization for further memory and latency gains.

- Through extensive experiments, we show that our approach is effective in both per-layer and cross-layer (shared dictionary) compression scenarios, consistently outperforming regular SVD, ASVD (Yuan et al., 2023), SVD-LLM (Wang et al., 2025a), and Basis Sharing (Wang et al., 2024) strategies as well as state-of-the-art structure-pruning methods (Ma et al., 2023; Shopkhoev et al., 2025).

## 2 RELATED WORK

Research on Transformer compression covers pruning, quantization, distillation, and matrix factorization. Below we provide a brief overview of the recent progress in these directions.

Early work on **pruning** showed substantial redundancy in deep nets (Han et al., 2015; Frankle & Carbin, 2019; Sanh et al., 2020). For LLMs, SparseGPT performs one-shot post-training pruning and achieves high sparsity with small quality drop (Frantar & Alistarh, 2023). Wanda reduces overhead further with simple activation-based rules (Sun et al., 2023). LLM-Pruner removes blocks using gradient-guided criteria and recovers accuracy with brief adapter tuning (Ma et al., 2023), while ReplaceMe substitutes multiple transformer blocks with a single linear transformation (Shopkhoev et al., 2025).

**Quantization** reduces precision to reduce memory consumption and accelerate inference. For weight-only post-training quantization, LLM.int8 enables INT8 inference via vector-wise quantization with a path for outliers (Dettmers et al., 2022); GPTQ/OPTQ minimize output error using Hessian to reach 3–4 bit weights (Frantar et al., 2022; 2023); AWQ derives scales from activations and protects a small set of salient weights (Lin et al., 2024), and SpQR stores outliers at higher precision to maintain quality at 3–4 bits (Dettmers et al., 2024). For quantizing both weights and activations, SmoothQuant rebalances channel ranges to achieve accurate W8A8 across matrix multiplications (Xiao et al., 2023), while QuaRot applies orthogonal rotations to suppress outliers, enabling end-to-end 4-bit inference including the KV cache (Ashkboos et al., 2024).

The general framework on **knowledge distillation** by Hinton et al. and sequence-level distillation by Kim and Rush established strong baselines (Hinton et al., 2015; Kim & Rush, 2016). DistilBERT demonstrates task-agnostic compression for Transformers (Sanh et al., 2019); TinyBERT uses a two-stage distillation procedure (Jiao et al., 2020); MobileBERT introduces a teacher-guided compact architecture (Sun et al., 2020); Patient-KD leverages multi-layer hints (Sun et al., 2019); MiniLM and MiniLMv2 distill self-attention relations (Wang et al., 2020; 2021). BERT-of-Theseus compresses models via progressive module replacement (Xu et al., 2020). Orca and Orca 2 show that

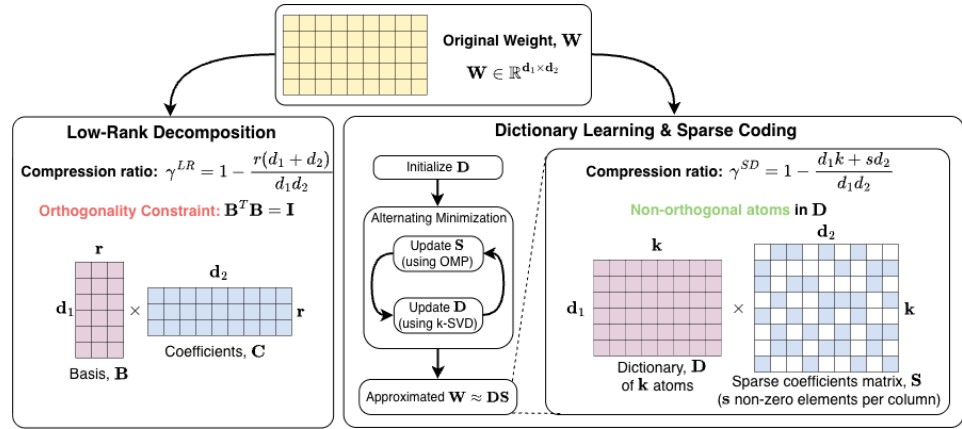

Figure 1: *Left side:* weight factorization methods using low-rank decomposition. Low-rank approximation decomposes a matrix into two dense matrices of lower rank. *Right side:* proposed CoSpaDi. A dictionary of $k$ atoms and a column-sparse coefficient matrix are employed. No restrictions on size of $k$ (undercomplete : $k < d_1$, complete: $k = d_1$ or overcomplete : $k > d_1$ dictionaries are possible), while sparsity is defined by $s$ non-zero elements per column of the coefficient matrix.

richer teacher signals such as explanations can improve reasoning in smaller students (Mukherjee et al., 2023; Mitra et al., 2023).

Classic **low-rank approximations** reduce parameters and FLOPs with limited loss (Denton et al., 2014). DRONE considers an objective related to the output activation for the weight approximation, (Chen et al., 2021). FWSVD introduces Fisher-weighted reconstruction to emphasize important directions (Hsu et al., 2022), while ASVD adapts truncation using activation-aware transforms and layer sensitivity (Yuan et al., 2023). SVD-LLM utilizes truncation-aware whitening , while SVD-LLM v2 optimizes budget allocation across layers (Wang et al., 2025a;b). Basis Sharing reuses one part of low-rank factorization across layers and learns layer-specific coefficients to exploit inter-layer redundancy (Wang et al., 2024).

**Dictionary learning and sparse coding.** Dictionary learning factorizes original weight into dictionary and sparse codes, yielding a union-of-subspaces model with strong results in vision and image compression (Engan et al., 1999; Aharon et al., 2006; Mairal et al., 2010; Gregor & LeCun, 2010; Elad, 2010). In NLP, GroupReduce explores block-wise low-rank/dictionary-style compression (Chen et al., 2018), and tensorized Transformers factorize attention and embeddings (Ma et al., 2019). Recent work also targets the KV-cache by learning universal dictionaries and decoding with OMP during inference (Kim et al., 2024). Complementary to these directions, a recent work proposes cross-layer weight sharing via a matrix-based dictionary learning formulation for Transformer attention, directly exploiting inter-layer redundancy (Zhussip et al., 2025). Cross-layer/shared-basis schemes in attention are thus closely related (Wang et al., 2024). A comprehensive, training-free approach to weight-space dictionary learning for LLMs — at both per-layer and cross-layer levels — remains underexplored and is the focus of this work.

## 3 PROPOSED METHOD

In this section, we first provide a conceptual link between low-rank weight approximation and dictionary learning. Then, we introduce the proposed Sparse Dictionary Learning (SDL) strategy for LLM compression covering all practical aspects including data-aware SDL, dictionary-sharing among layers, compression ratios and inference complexity.

### 3.1 LOW-RANK WEIGHT APPROXIMATION

A widely adopted strategy to reduce the parameter count in LLMs is to approximate weight matrices $\mathbf{W} \in \mathbb{R}^{d_1 \times d_2}$ — representing the parameters of network layers — with matrices of reduced rank

$r < \min(d_1, d_2)$. Such a low-rank approximation can be derived as follows:

$$\tilde{\mathbf{W}}^\star = \underset{\operatorname{rank}(\tilde{\mathbf{W}})=r}{\arg\min} \left\| \mathbf{W} - \tilde{\mathbf{W}} \right\|_F. \tag{1}$$

According to the Eckart–Young–Mirsky theorem (Eckart & Young, 1936), the orthogonal projection to the space of $r$-rank matrices admits an analytical solution. Specifically, if $\mathbf{W}$ admits the singular value decomposition $\mathbf{W} = \mathbf{U}\boldsymbol{\Sigma}\mathbf{V}^\mathsf{T}$, with $\mathbf{U} \in \mathbb{R}^{d_1 \times k}$, $\boldsymbol{\Sigma} \in \mathbb{R}^{k \times k}$ and $\mathbf{V} \in \mathbb{R}^{d_2 \times k}$ with $k = \min(d_1, d_2)$, then the minimizer of Eq. (1) can be expressed as: $\tilde{\mathbf{W}}^\star = \mathbf{U}_r \boldsymbol{\Sigma}_r \mathbf{V}_r^\mathsf{T}$, where $\mathbf{U}_r \in \mathbb{R}^{d_1 \times r}$ contains the first $r$ left singular vectors, $\boldsymbol{\Sigma}_r \in \mathbb{R}^{r \times r}$ holds the top-$r$ singular values, and $\mathbf{V}_r \in \mathbb{R}^{d_2 \times r}$ contains the first $r$ right singular vectors of $\mathbf{W}$, respectively.

There is a deep connection between this problem and the principal component analysis (PCA) (Bishop & Nasrabadi, 2006). Specifically, consider the weight matrix $\mathbf{W}$ as a collection of $d_1$-dimensional vectors $\mathbf{W} = [\mathbf{w}_1, \quad \ldots, \mathbf{w}_{d_2}]$, where $\boldsymbol{w}_j \in \mathbb{R}^{d_1}$, with $j = 1, \ldots, d_2$. We seek to approximate each vector $\mathbf{w}_j$ as a linear combination of basis vectors spanning a lower-dimensional subspace of $\mathbb{R}^{d_1}$. The optimal basis and coefficients can be found by minimizing the total approximation error:

$$\mathcal{J} = \sum_{j=1}^{d_2} \left\| \mathbf{w}_j - \sum_{i=1}^{r} c_{i,j} \boldsymbol{b}_i \right\|_2^2 \equiv \| \mathbf{W} - \mathbf{BC} \|_F^2, \tag{2}$$

subject to the orthogonality constraint $\mathbf{B}^\mathsf{T}\mathbf{B} = \mathbf{I}$, where $\mathbf{B} = [\boldsymbol{b}_1, \ldots, \boldsymbol{b}_r] \in \mathbb{R}^{d_1 \times r}$ is the basis matrix, $\mathbf{C} \in \mathbb{R}^{r \times d_2}$ is the coefficient matrix with entries $c_{i,j}$, and $\mathbf{I} \in \mathbb{R}^{r \times r}$ is the identity matrix.

The optimal pair $(\mathbf{B}^*, \mathbf{C}^*)$ is obtained by solving:

$$\mathbf{B}^*, \mathbf{C}^* = \underset{\mathbf{B}, \mathbf{C}}{\arg\min} \, \mathcal{J} \text{ s.t } \mathbf{B}^\mathsf{T}\mathbf{B} = \mathbf{I}. \tag{3}$$

The solution is given by $\mathbf{B}^* = \mathbf{U}_r$ and $\mathbf{C}^* = \boldsymbol{\Sigma}_r \mathbf{V}_r^\mathsf{T}$ (we refer to Appendix A.1 for the proof). This result highlights that in the low-rank factorization $\mathbf{W} \approx \mathbf{BC}$, the matrix $\mathbf{B}$ corresponds to the basis of the shared $r$-dimensional subspace in which the columns of $\mathbf{W}$ approximately lie, and $\mathbf{C}$ contains their coordinate vectors (coefficients) with respect to that basis.

## 3.2 SPARSE DICTIONARY LEARNING

Motivated by this interpretation, we propose an alternative strategy for compressing the weights $\mathbf{W}$, based on the sparse dictionary learning methodology. Specifically, rather than modeling each column of the weight matrix as a linear combination of basis vectors $\boldsymbol{b}_i$, we aim to learn a dictionary $\mathbf{D} \in \mathbb{R}^{d_1 \times k}$ and approximate each column $\mathbf{w}_j$ as a linear combination of dictionary atoms $\mathbf{d}_i \in \mathbb{R}^{d_1}$ with $i = 1, \ldots, k$.

Although the two strategies appear similar, they differ in two key respects. First, in dictionary learning, unlike low-rank approximation, only a *subset* of atoms is used to represent each weight vector $\mathbf{w}_j$. This principle, known in the literature as *sparse coding* (Lee et al., 2006), is motivated by the observation that learned dictionaries typically yield sparse representations: each signal (column) activates only a few atoms. In other words, instead of approximating the weight matrix $\mathbf{W}$ as the product of two dense matrices, the orthogonal basis $\mathbf{B}$ and the coefficient matrix $\mathbf{C}$, we approximate it as the product of a dense dictionary matrix $\mathbf{D}$ and a *column-wise sparse* coefficient matrix $\mathbf{S}$, where each column $\mathbf{s}_j$ indicates which atoms contribute to the representation of $\mathbf{w}_j$, and with what weights. Second, and of significant practical consequence, we do not impose orthogonality constraints on the dictionary atoms. This grants greater representational flexibility, allowing the model to adapt more effectively to the intrinsic structure of the weight matrix.

Finally, under the proposed strategy, rather than enforcing a *shared* low-dimensional subspace for all columns, each weight vector $\mathbf{w}_j$ is represented within its own *low-dimensional subspace* spanned by the atoms it activates. Consequently, the entire matrix $\mathbf{W}$ is modeled as a *union of low-dimensional subspaces*, introducing additional representational capacity and enabling lower approximation error compared to a single shared subspace.

Formally, the dictionary learning problem can be expressed as:

$$\mathbf{D}^\star, \mathbf{S}^\star = \arg\min_{\mathbf{D},\mathbf{S}} \|\mathbf{W} - \mathbf{DS}\|_F^2 \quad \text{s.t.} \quad \|\mathbf{s}_j\|_0 \leq s \quad \forall j = 1, \ldots, d_2, \tag{4}$$

where $\mathbf{s}_j$ denotes the $j$-th column of the sparse coefficient matrix $\mathbf{S}$, and $\|\cdot\|_0$ denotes the $\ell_0$ pseudo-norm, which counts the number of nonzero entries in a vector.

The optimization problem in Eq. (4) is NP-hard in its original form and admits no closed-form solution. Nevertheless, it has been extensively studied, and several efficient algorithms have been proposed to obtain high-quality approximate solutions. Among these, K-SVD (Aharon et al., 2006) is a well-established algorithm widely adopted across diverse applications.

K-SVD alternates between two main steps: (a) **Sparse coding**: Each signal $\mathbf{w}_j$ is approximated as a sparse linear combination of dictionary atoms (typically via orthogonal matching pursuit or LASSO). (b) **Dictionary update**: Each atom $\mathbf{d}_i$ is refined to better fit the data while preserving the sparsity pattern in $\mathbf{S}$. This is performed sequentially per atom: the algorithm identifies the subset of signals that activate $\mathbf{d}_i$, computes the residual error over those signals, and applies a rank-1 SVD to jointly update the atom and its associated coefficients.

### 3.3 CoSpaDi: Activation-Aware SDL

In several recent works, including (Chen et al., 2021), it has been observed that LLM weight matrices generally do not exhibit intrinsic low-rank structure. Consequently, direct low-rank approximation of the weights often leads to significant performance degradation. However, under the hypothesis that the input latent features to a given layer reside in a low-dimensional subspace, it remains feasible and often effective to approximate the corresponding layer weights as low-rank. This approach is well-motivated: the goal is not to preserve the weights in isolation, but rather to maintain the fidelity of the *activations* they produce when applied to low-dimensional inputs.

Within this framework, an effective approximation of $\mathbf{W}$ can be obtained by minimizing the output error:

$$\tilde{\mathbf{W}}^* = \arg\min_{\tilde{\mathbf{W}}} \left\|\mathbf{XW} - \mathbf{X}\tilde{\mathbf{W}}\right\|_F \quad \text{s.t.} \quad \text{rank}(\mathbf{X}\tilde{\mathbf{W}}) = r, \tag{5}$$

where $\mathbf{X} \in \mathbb{R}^{N \times d_1}$ is a matrix of $N$ calibration input vectors $\left\{\mathbf{x}_i \in \mathbb{R}^{d_1}\right\}_{i=1}^N$. This minimization admits the analytical solution: $\tilde{\mathbf{W}} = \mathbf{L}^{-1}\mathbf{U}_r\mathbf{\Sigma}_r\mathbf{V}_r^\top = \mathbf{BC}$, where $\mathbf{L} \in \mathbb{R}^{d_1 \times d_1}$ is a non-singular transformation matrix derived from the calibration data (see Appendix A.2 for the derivation).

Motivated by this data-aware perspective, we propose adapting our sparse dictionary learning framework accordingly. Rather than directly approximating $\mathbf{W}$ as $\tilde{\mathbf{W}} = \mathbf{DS}$, we instead minimize the reconstruction error of the *output activations* $\mathbf{Z} = \mathbf{XW}$:

$$\mathbf{D}^\star, \mathbf{S}^\star = \arg\min_{\mathbf{D},\mathbf{S}} \|\mathbf{XW} - \mathbf{XDS}\|_F^2 \quad \text{s.t.} \quad \|\mathbf{s}_j\|_0 \leq s \quad \forall j = 1, \ldots, d_2. \tag{6}$$

To simplify this optimization, let $\mathbf{Y} = \mathbf{XL}^{-1} \in \mathbb{R}^{N \times d_1}$ be a column-orthogonal matrix, i.e., $\mathbf{Y}^\top\mathbf{Y} = \mathbf{I}_{d_1}$, obtained by applying a linear transformation $\mathbf{L} \in \mathbb{R}^{d_1 \times d_1}$ to $\mathbf{X}$. We assume $N \geq d_1$ and that $\mathbf{X}$ has full column rank — conditions typically satisfied with sufficient calibration data. The matrix $\mathbf{L}$ can be computed via QR decomposition, SVD of $\mathbf{X}$, or Cholesky/eigen-decomposition of $\mathbf{X}^\top\mathbf{X}$.

Introducing the auxiliary variables $\mathbf{W}_L = \mathbf{LW}$ and $\mathbf{D}_L = \mathbf{LD}$, and noting that the Frobenius norm is invariant under left-multiplication by a column-orthogonal matrix, we can reformulate Eq. (6) as:

$$\mathbf{D}_L^\star, \mathbf{S}^\star = \arg\min_{\mathbf{D}_L,\mathbf{S}} \|\mathbf{W}_L - \mathbf{D}_L\mathbf{S}\|_F^2 \quad \text{s.t.} \quad \|\mathbf{s}_j\|_0 \leq s \quad \forall j = 1, \ldots, d_2. \tag{7}$$

Since $\mathbf{L}$ is invertible, the final structured approximation of the original weights is given by:

$$\tilde{\mathbf{W}} = \mathbf{D}_a\mathbf{S}, \quad \text{where} \quad \mathbf{D}_a = \mathbf{L}^{-1}\mathbf{D}_L \tag{8}$$

is the *activation-aware learned dictionary*. Pseudocode for the full procedure is provided in Appendix A.3.

**Cross-Layer SDL**   Although most compression techniques focus on intra-layer optimizations, the repetitive layered structure of LLMs suggests the presence of significant *inter-layer redundancy*. To exploit this, we propose sharing a single dictionary across weight matrices of the same type from multiple layers. Specifically, let $\mathcal{G}$ denote a group of layer indices (e.g., all attention projection layers or all FFN layers). We form a grouped weight matrix by horizontally concatenating the corresponding layer weights: $\mathbf{W}_G = [\mathbf{W}_{\ell_1}, \quad \mathbf{W}_{\ell_2}, \quad \ldots, \quad \mathbf{W}_{\ell_L}] \in \mathbb{R}^{d_1 \times (d_2 \cdot L)}$, where each $\mathbf{W}_\ell \in \mathbb{R}^{d_1 \times d_2}$, and $L = |\mathcal{G}|$ is the number of layers in the group. We then apply the same activation-aware dictionary learning procedure as before, but now to $\mathbf{W}_G$. To ensure the approximation remains data-aware across all layers in the group, we construct a grouped calibration matrix by vertically stacking the corresponding input batches: $\mathbf{X}_G = [\mathbf{X}_{\ell_1}; \quad \mathbf{X}_{\ell_2}; \quad \ldots; \quad \mathbf{X}_{\ell_L}] \in \mathbb{R}^{(N \cdot L) \times d_1}$ where each $\mathbf{X}_\ell \in \mathbb{R}^{N \times d_1}$ is the calibration input for layer $\ell$. The transformation matrix $\mathbf{L} \in \mathbb{R}^{d_1 \times d_1}$ is then computed from $\mathbf{X}_G$ (e.g., via QR or Cholesky decomposition), ensuring that $\mathbf{Y}_G = \mathbf{X}_G \mathbf{L}^{-1}$ is column-orthogonal. The optimization proceeds by solving Eq. (7) with $\mathbf{W}_L = \mathbf{L}\mathbf{W}_G$, yielding a shared dictionary $\mathbf{D}_a$ and a sparse coefficient matrix $\mathbf{S}_G \in \mathbb{R}^{k \times (d_2 \cdot L)}$.

This approach, inspired by (Wang et al., 2024), reduces memory overhead by amortizing the dictionary cost across multiple layers, while preserving activation fidelity through data-aware calibration. Each layer's compressed weights are then recovered by slicing the corresponding block from $\tilde{\mathbf{W}}_G = \mathbf{D}_a \mathbf{S}_G$.

**Compression ratio.**   For the low-rank setting, the compression ratio is $\gamma^{\mathrm{LR}} := 1 - \frac{r(d_1 + d_2)}{d_1 d_2}$, where $r$ is the retained rank. For CoSpaDi, we store a dictionary and sparse codes plus a binary mask (details are provided in appendix A.4), giving

$$\gamma^{\mathrm{SD}} := 1 - \frac{\overbrace{d_1 k}^{\text{dict. (bf16)}} + \overbrace{s d_2}^{\text{codes (bf16)}} + \overbrace{(k d_2)/16}^{\text{mask (1 bit/entry)}}}{d_1 d_2}. \tag{9}$$

Unlike the low-rank case, $\gamma^{\mathrm{SD}}$ depends on two knobs — the number of atoms $k$ and sparsity $s$ — allowing us to trade off model capacity and storage at fixed budget. We parameterize this trade-off by the ratio $\rho := k/s$, which uniquely determines $(k, s)$ at a target $\gamma^{\mathrm{SD}}$:

$$k = \frac{(1 - \gamma^{\mathrm{SD}}) \, d_1 d_2}{d_1 + \frac{d_2}{\rho} + \frac{d_2}{16}}, \qquad s = \frac{k}{\rho}. \tag{10}$$

**Inference Complexity**   In terms of computational efficiency, low-rank and dictionary learning exhibit distinct inference-time complexity profiles: the former has a cost of $Tr(d_1 + d_2)$, whereas the latter – when exploiting sparsity and reusing inner products – achieves $T d_1 K_{\mathrm{active}} + T s d_2$ (see appendix A.5), potentially yielding superior efficiency under favorable sparsity patterns. Although both methods share identical theoretical complexity under matched compression ratios, practical inference latency varies significantly due to factors such as the number of active atoms, indexing overhead, and hardware-specific kernel efficiency. Further details regarding complexity derivations are provided in Appendix A.5.

## 4 EXPERIMENTS

In this section, we present our experimental setup. We first compare CoSpaDi with low-rank baselines in the *per-layer* setting, where each linear weight matrix is compressed independently. We then investigate cross-layer sharing, analyze the impact of coefficient matrix quantization, and conduct ablation studies on the $ks$-ratio.

### 4.1 EXPERIMENTAL SETUP

We evaluate our method in both per-layer and cross-layer settings. For per-layer evaluations, we consider LLaMA-3.2 1B, Qwen-3 0.6B, LLaMA-3 8B, and Qwen-3 8B. For the cross-layer study, we follow the Basis Sharing (Wang et al., 2024) setup and conduct experiments on LLaMA-2 7B. All models are evaluated in a zero-shot setting on PIQA (Bisk et al., 2019), HellaSwag (Zellers et al., 2019), OpenAI LAMBADA (Paperno et al., 2016), ARC-easy and ARC-challenge (Clark et al., 2018), SciQ (Welbl et al., 2017), Race (Lai et al., 2017), and MMLU (Hendrycks et al., 2021), while perplexity is additionally reported on WikiText (Merity et al., 2016) and LAMBADA-OpenAI. Compression is applied with target ratios of 0.2-0.5 with 0.1 step. For methods requiring calibration

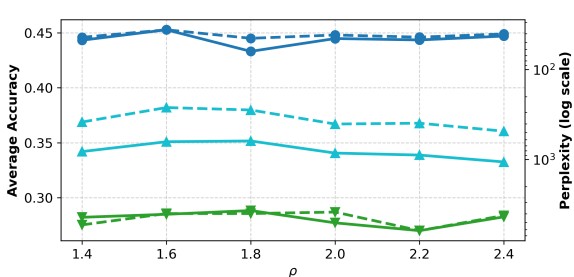

Figure 2: Dual-axis plot showing average accuracy (—— solid lines, left axis) and perplexity (– – – dashed lines, right axis, logarithmic scale with inverted direction) as functions of $\rho$ for Llama3.2-1B under three compression levels: 0.2, 0.3 and 0.4. Perplexity decreases upward due to axis inversion.

| CR | Bitwidth | Avg. Acc. | PPL |
|------|----------|-----------|-------|
| 0.17 | bFP16 | 62.0 | 19.4 |
| 0.18 | bFP15 | 62.0 | 19.5 |
| 0.20 | bFP14 | 61.8 | 19.7 |
| 0.27 | bFP16 | 54.1 | 43.4 |
| 0.29 | bFP15 | 54.0 | 43.7 |
| 0.30 | bFP14 | 53.7 | 44.6 |
| 0.38 | bFP16 | 41.0 | 180.1 |
| 0.39 | bFP15 | 40.9 | 181.8 |
| 0.40 | bFP14 | 40.7 | 184.1 |
| 0.48 | bFP16 | 28.5 | 822.6 |
| 0.49 | bFP15 | 28.4 | 815.0 |
| 0.50 | bFP14 | 28.4 | 809.6 |

Table 1: Results on truncation of bfloat16 mantissa bits of coefficient matrix with reported average accuracy (Avg. Acc.) and Wiki Text word perplexity (PPL) for Llama3-8B resulting in different compression ratios (CR).

data, we randomly sample 256 sequences from the RefinedWeb dataset (Penedo et al., 2023), each consisting of 1024 tokens. Unless specified otherwise, we compress all dense linear projections in self-attention (Q/K/V/O) and feed-forward network (gate/up/down). Embeddings and lm_head are left intact.

For low-rank methods the rank is uniquely defined with $\gamma^{LR}$ and we floored it to the nearest integer. For CoSpaDi we fixed $\rho = 2$ ($k/s$ ratio) for all experiments, so $k$ and $s$ were obtained according to Eq. (10) and then were floored to the nearest integers.

In CoSpaDi we employ K-SVD using power iterations instead of full-svd for the dictionary updates. Specifically, we use 60 K-SVD iterations and 8 power iterations to ensure stable convergence. Further implementation details and convergence analysis are provided in Appendix A.8.

## 4.2 ABLATION STUDY

**Influence of $k/s$-ratio.** In the proposed CoSpaDi we can redistribute capacity across both $k$ and $s$ while preserving predefined compression ratio by varying the $\rho$ value ($k/s$). We verified its influence on the Llama 3.2 1B and report the metrics in Fig. 2.

From the provided plots we can observe that depending on the compression ratio the optimal $ks$-ratio differs, thus, for simplicity in further experiments we select in all cases $\rho = 2$.

**Data-Free and Data-Aware Scenarios** We argue that our proposed CoSpaDi allows for a more flexible representation compared to low-rank approximation, regardless of whether data-free or data-aware scenarios are considered. To prove this claim we performed an ablation study on Llama3-1B for different compression levels. We selected SVD for the data-free scenario and SVD-LLM for the data-aware case, while varying the compression ratio from 0.2 to 0.5. In Table 2 we report the results only for 0.2-0.5 compression. These results clearly indicate that in both data-free and data-aware settings the SDL-based methods outperform low-rank baselines by a wide margin, while CoSpaDi outperforms all methods.

**Quantization of Sparse Coefficient Matrix** We investigated the quantization of the coefficient matrix $\mathbf{S}$ in the post-training regime to leverage the memory add-on due to the requirement of storing the indices of the nonzero values. We utilized a naive mantissa bit truncation of the original bfloat16 coefficient values. In Table 1 we report how truncation affects the performance at different compression levels. According to the obtained results, truncation of 2 bits leads to negligible drop. Thus, in all our experiments CoSpaDi consists of bfloat16 dictionary and coefficient matrices but is evaluated with the truncated version of the sparse coefficient matrix using 14-bits.

## 4.3 MAIN RESULTS

**Per-Layer Scenario:** In this section we apply CoSpaDi for each projection layer independently and compare it against current sota training-free low-rank method – SVD-LLM (Wang et al., 2025a).

Table 2: SDL-based methods comparison vs low-rank counterparts in data-free and data-aware scenarios on Llama3.2-1B at different compression ratios (CR). We denote CoSpaDi† as the proposed method without using calibration data. Best results are provided in **bold**.

| Method | Data-Aware | CR | PIQA | Hella Swag | LAMBADA | ARC-e | ARC-c | SciQ | Race | MMLU | Avg. | Wiki Text | LAMBADA |
|---|---|---|---|---|---|---|---|---|---|---|---|---|---|
| | | | | | | Accuracy↑ | | | | | | Perplexity↓ | |
| **Llama3.2 1B** | – | – | 74.5 | 63.7 | 63.0 | 60.5 | 36.2 | 88.3 | 37.8 | 37.0 | 57.6 | 11.6E+00 | 5.7E+00 |
| SVD | ✗ | | 52.0 | 25.7 | 0.0 | 24.7 | 22.1 | 20.0 | 21.4 | **25.4** | 23.9 | 2.9E+06 | 4.6E+06 |
| CoSpaDi† | ✗ | 0.2 | 51.7 | 26.4 | 0.0 | 25.4 | 25.3 | 21.0 | 21.8 | 24.1 | 24.5 | 3.3E+05 | 2.2E+06 |
| SVD-LLM | ✓ | | 62.1 | 36.4 | 24.4 | 36.0 | 25.1 | 64.9 | 29.0 | 23.0 | 37.6 | 1.7E+02 | 1.7E+02 |
| CoSpaDi | ✓ | | **66.1** | **42.9** | **38.4** | **39.9** | **26.0** | **71.6** | **31.7** | 24.8 | **42.7** | **6.4E+01** | **3.5E+01** |
| SVD | ✗ | | 52.3 | 25.6 | 0.0 | 24.2 | 23.5 | 19.5 | 21.9 | **27.0** | 24.3 | 1.1E+06 | 3.9E+06 |
| CoSpaDi† | ✗ | 0.3 | 50.5 | 26.3 | 0.0 | 24.5 | **26.1** | 21.8 | 21.5 | 25.5 | 24.5 | 2.1E+05 | 4.3E+06 |
| SVD-LLM | ✓ | | 55.7 | 30.1 | 9.1 | 30.5 | 21.5 | 45.9 | 25.8 | 23.2 | 30.2 | 5.9E+02 | 2.5E+03 |
| CoSpaDi | ✓ | | **56.9** | **32.4** | **18.2** | **31.9** | 22.1 | **56.7** | **28.0** | 23.1 | **33.7** | **2.9E+02** | **6.6E+02** |
| SVD | ✗ | | 52.8 | 25.9 | 0.0 | 23.9 | 21.3 | 19.9 | 22.2 | **26.9** | 24.1 | 1.2E+06 | 4.2E+06 |
| CoSpaDi† | ✗ | 0.4 | 51.0 | 26.3 | 0.0 | 25.5 | **26.9** | 21.3 | 21.2 | 25.5 | 24.7 | 3.1E+06 | 3.7E+07 |
| SVD-LLM | ✓ | | 51.8 | 27.3 | 1.3 | 26.9 | 22.9 | 32.3 | **24.4** | 23.0 | 26.2 | 1.6E+03 | 3.3E+04 |
| CoSpaDi | ✓ | | **53.5** | **28.2** | **3.8** | **27.8** | 23.0 | **36.9** | 24.0 | 23.1 | **27.5** | **8.0E+02** | **9.2E+03** |
| SVD | ✗ | | **53.9** | 25.7 | 0.0 | 24.4 | 20.4 | 20.4 | 20.8 | 26.9 | 24.1 | 1.2E+06 | 1.9E+07 |
| CoSpaDi† | ✗ | 0.5 | 51.2 | 26.1 | 0.0 | 25.7 | **26.5** | 22.8 | 22.1 | **27.0** | 25.2 | 6.1E+05 | 1.1E+07 |
| SVD-LLM | ✓ | | 51.1 | 26.6 | 0.0 | 26.1 | 25.9 | 26.1 | 23.9 | 23.0 | 25.4 | 3.1E+03 | 1.0E+05 |
| CoSpaDi | ✓ | | 51.7 | **27.0** | **0.3** | 26.3 | 24.0 | **29.5** | **24.2** | 23.3 | **25.8** | **1.8E+03** | 7.3E+04 |

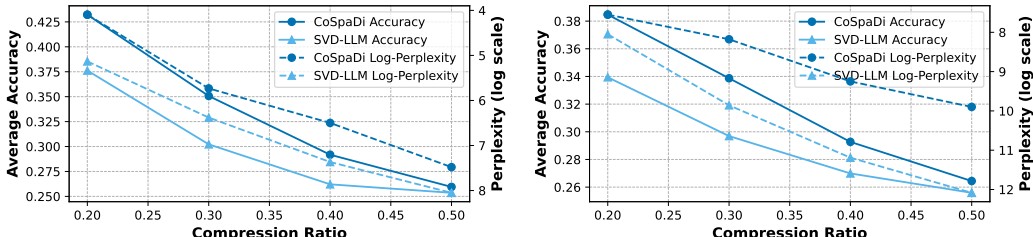

(a) Accuracy & Perplexity for LLaMA3.2-1B  (b) Accuracy & Perplexity for Qwen3-0.6B

Figure 3: Average benchmark accuracy and WikiText perplexity for (a) LLaMA-3.2-1B and (b) Qwen-3 0.6B using SVD-LLM and CoSpaDiwith respect to compression ratio.

From the provided plots CoSpaDi significantly outperforms SVD-LLM in both avg. accuracy and perplexity for small Llama3 and Qwen3 models. We further investigate how our method scales to larger models of the same families with 8B parameters. The related results are provided in Table 3.

We have also performed comparisons on the same models and compression levels against other training-free compression strategies. In particular we compare CoSpaDi against ASVD (Yuan et al., 2023), ReplaceMe (Shopkhoev et al., 2025) and LLMPruner (Ma et al., 2023), with the last two being sota pruning methods. These results are provided in Appendix A.6

**Cross-Layer Scenario**  In this section, we validate CoSpaDi on the cross-layer compression scenario, where a common dictionary is shared across different layers. To ensure a fair comparison, we adopt the identical layer selection and grouping procedure introduced in Basis Sharing (Wang et al., 2024) (for a detailed description we refer to Appendix A.7) and compare the performance of the two methods on the Llama2-7B model (Touvron et al., 2023b). As shown in Table 4, CoSpaDi consistently outperforms Basis Sharing by a large margin across all benchmarks and compression rates (0.2–0.5). This performance gap suggests that CoSpaDi's sparsity-aware, dictionary-based decomposition better preserves task-critical features under aggressive compression. We believe that a more sophisticated layer grouping strategy, based on feature map similarity or end-to-end differentiable search, could work better with CoSpaDi and further widen this performance margin, but we leave this as a future research direction. We provide details for $k$ (dictionary size) and $s$ (sparsity) parameters in the Appendix A.7.

**Limitations and Future Work**  In this work, we employ K-SVD as a representative instantiation of our framework, though other dictionary learning algorithms could be similarly applied. A key

Table 3: Performance comparison of CoSpaDi vs sota SVD-LLM on Llama3-8B and Qwen3-8B at different compression levels on different benchmarks. Best results are highlighted with **bold**.

| Method | CR | Accuracy↑ | | | | | | | | | Perplexity↓ | |
|---|---|---|---|---|---|---|---|---|---|---|---|---|
| | | PIQA | Hella Swag | LAMBADA | ARC-e | ARC-c | SciQ | Race | MMLU | Avg. | Wiki Text | LAMBADA |
| **Llama3 8B** | – | 80.7 | 79.1 | 75.6 | 77.7 | 53.5 | 93.9 | 40.3 | 62.2 | 70.4 | 7.3E+00 | 3.1E+00 |
| SVD-LLM | 0.2 | 71.1 | 58.4 | 59.3 | 55.5 | 34.0 | 86.4 | 35.5 | 32.6 | 54.1 | 4.1E+01 | 1.1E+01 |
| CoSpaDi | | **75.2** | **66.5** | **73.8** | **66.5** | **41.6** | **89.5** | **38.2** | **42.8** | **61.8** | **2.0E+01** | **4.3E+00** |
| SVD-LLM | 0.3 | 65.8 | 46.4 | 38.1 | 41.9 | 27.7 | 70.0 | 31.8 | 27.2 | 43.6 | 1.5E+02 | 6.1E+01 |
| CoSpaDi | | **70.5** | **56.2** | **61.3** | **54.2** | **33.5** | **85.7** | **36.2** | **32.2** | **53.7** | **4.5E+01** | **9.2E+00** |
| SVD-LLM | 0.4 | 60.3 | 34.5 | 11.4 | 32.4 | 24.5 | 44.2 | 25.7 | 23.1 | 32.0 | 5.5E+02 | 1.3E+03 |
| CoSpaDi | | **63.7** | **41.4** | **30.3** | **39.1** | **26.6** | **68.5** | **30.5** | **25.4** | **40.7** | **1.8E+02** | **1.2E+02** |
| SVD-LLM | 0.5 | 53.8 | 27.9 | 0.4 | 27.3 | 23.9 | 26.6 | 22.3 | 23.0 | 25.7 | 1.4E+03 | 2.8E+04 |
| CoSpaDi | | **56.4** | **31.0** | **3.9** | **29.9** | **22.2** | **37.7** | **22.9** | 23.0 | **28.4** | **8.1E+02** | **7.6E+03** |
| **Qwen3 8B** | – | 77.7 | 74.9 | 64.1 | 80.7 | 56.7 | 95.7 | 40.9 | 73.0 | 70.5 | 1.2E+01 | 4.6E+00 |
| SVD-LLM | 0.2 | 73.8 | 63.9 | 62.2 | 68.7 | 45.7 | 90.1 | 40.5 | 54.7 | 62.5 | 2.1E+01 | 6.4E+00 |
| CoSpaDi | | **76.5** | **68.0** | **65.6** | **72.2** | **48.9** | **93.2** | **40.7** | **60.8** | **65.7** | **1.8E+01** | **4.9E+00** |
| SVD-LLM | 0.3 | 70.4 | 55.2 | 53.8 | 59.3 | 37.1 | 87.2 | 38.4 | 44.8 | 55.8 | 2.7E+01 | 1.1E+01 |
| CoSpaDi | | **72.4** | **60.5** | **62.6** | **63.9** | **41.2** | **88.4** | **39.5** | **51.3** | 59.97 | **2.3E+01** | **6.3E+00** |
| SVD-LLM | 0.4 | 66.3 | 44.6 | 37.9 | 45.0 | 28.1 | 77.3 | 35.3 | 29.1 | 45.4 | 4.3E+01 | 3.6E+01 |
| CoSpaDi | | **68.9** | **49.0** | **49.9** | **49.4** | **29.9** | **82.0** | **36.8** | **36.6** | **50.3** | **3.6E+01** | **1.5E+01** |
| SVD-LLM | 0.5 | 60.2 | 35.4 | 19.6 | 33.6 | 22.8 | 66.7 | 29.1 | 23.1 | 36.3 | 8.7E+01 | 2.7E+02 |
| CoSpaDi | | **61.6** | **38.0** | **27.9** | **35.7** | **23.0** | **68.6** | **31.7** | 23.1 | **38.7** | **6.8E+01** | **1.1E+02** |

Table 4: Performance comparison of CoSpaDi vs Basis Sharing (Wang et al., 2024) on Llama2-7B under different compression levels on various benchmarks. Best results are highlighted with **bold**.

| Method | CR | Accuracy↑ | | | | | | | | | Perplexity↓ | |
|---|---|---|---|---|---|---|---|---|---|---|---|---|
| | | PIQA | Hella Swag | LAMBADA | ARC-e | ARC-c | SciQ | Race | MMLU | Avg. | Wiki Text | LAMBADA |
| **Llama2 7B** | – | 78.0 | 57.1 | 73.7 | 76.1 | 43.3 | 93.9 | 39.2 | 40.9 | 62.8 | 8.7 | 3.4 |
| Basis Sharing | 0.2 | 70.1 | 43.4 | 63.0 | 66.5 | 33.7 | **91.0** | 34.9 | 24.8 | 53.4 | 15.2 | 7.2 |
| CoSpaDi | | **74.6** | **66.3** | **71.3** | **67.4** | **39.3** | 88.8 | **38.3** | **27.2** | **59.2** | **11.7** | **4.3** |
| Basis Sharing | 0.3 | 65.6 | 37.6 | 53.5 | 58.8 | 27.3 | 86.8 | 32.6 | 23.2 | 48.2 | 22.2 | 13.6 |
| CoSpaDi | | **70.8** | **58.5** | **64.5** | **61.9** | **34.3** | **87.6** | **35.9** | **24.0** | **54.7** | **15.4** | **6.3** |
| Basis Sharing | 0.4 | 60.6 | 33.1 | 41.1 | 49.5 | 23.6 | **83.0** | 29.7 | **23.1** | 42.95 | 39.5 | 37.7 |
| CoSpaDi | | **64.3** | **48.0** | **51.8** | **52.3** | **29.9** | 81.5 | **33.2** | 23.0 | **48.0** | **25.2** | **14.6** |
| Basis Sharing | 0.5 | 56.3 | 29.8 | 23.8 | 37.3 | 20.5 | 74.5 | 25.4 | 22.8 | 36.3 | 98.6 | 225.1 |
| CoSpaDi | | **57.5** | **36.6** | **32.6** | **41.7** | **24.9** | **74.7** | **26.3** | **22.9** | **39.7** | **57.3** | **72.6** |

limitation of k-SVD lies in its reliance on Orthogonal Matching Pursuit (OMP) for sparse coding and its sequential atom updates, which can be computationally slow. However, more efficient variants do exist that accelerate convergence while maintaining the same performance and we plan to explore them in the future. As an additional direction for future work, we plan to investigate adaptive capacity allocation across the model by dynamically distributing sparsity and dictionary size according to layer-specific demands, while leveraging cross-layer dictionary sharing in a structured and scalable manner.

## 5 CONCLUSIONS

In summary, we introduced **CoSpaDi**, a novel training-free compression framework for LLMs that challenges the dominance of low-rank approximation in post-training model compression. By introducing dictionary learning with sparse coding as a more expressive alternative, we shift from the rigid constraint of a shared linear subspace to a flexible union-of-subspaces representation, where each column of a weight matrix is approximated using a sparse combination of learned dictionary atoms. This work demonstrates that moving beyond SVD-driven paradigms, long considered the default for matrix compression in LLMs, can yield significant gains in model fidelity under aggressive compression. We hope **CoSpaDi** serves as a conceptual stepping stone toward richer, data-aware factorizations inspired by classical signal processing, yet tailored for the complexity of modern language models.

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

# A    APPENDIX

## A.1    DERIVATION OF THE OPTIMAL PAIR OF BASIS AND COEFFICIENT MATRICES

We are seeking the optimal pair $(\mathbf{B}^\star, \mathbf{C}^\star)$ that minimizes the constrained problem in Eq. (3). First, we rewrite the objective $\mathcal{J}$ in its equivalent form:

$$\mathcal{J} = \mathrm{tr}\left(\mathbf{W}^\mathsf{T}\mathbf{W}\right) - 2\,\mathrm{tr}\left(\mathbf{W}^\mathsf{T}\mathbf{BC}\right) + \mathrm{tr}\left(\mathbf{C}^\mathsf{T}\mathbf{B}^\mathsf{T}\mathbf{BC}\right). \tag{11}$$

Next, we consider the basis matrix $\mathbf{B}$ as fixed and compute the gradient of the objective w.r.t the matrix coefficient $\mathbf{C}$ as

$$\nabla_{\mathbf{C}}\mathcal{J} = 2\mathbf{B}^\mathsf{T}\mathbf{BC} - 2\mathbf{B}^\mathsf{T}\mathbf{W}. \tag{12}$$

Setting the gradient to zero and taking into account the constraint $\mathbf{B}^\mathsf{T}\mathbf{B} = \mathbf{I}$, we can recover the optimum matrix coefficient as: $\mathbf{C} = \mathbf{B}^\mathsf{T}\mathbf{W}$. Now, putting back $\mathbf{C}$ in Eq. (11) we get:

$$\begin{aligned}
\mathcal{J} &= \mathrm{tr}\left(\mathbf{W}^\mathsf{T}\mathbf{W}\right) - 2\,\mathrm{tr}\left(\mathbf{W}^\mathsf{T}\mathbf{BB}^\mathsf{T}\mathbf{W}\right) + \mathrm{tr}\left(\mathbf{W}^\mathsf{T}\mathbf{BB}^\mathsf{T}\mathbf{BB}^\mathsf{T}\mathbf{W}\right) \\
&= \mathrm{tr}\left(\mathbf{W}^\mathsf{T}\mathbf{W}\right) - \mathrm{tr}\left(\mathbf{W}^\mathsf{T}\mathbf{BB}^\mathsf{T}\mathbf{W}\right) \ \text{(from the constraint } \mathbf{B}^\mathsf{T}\mathbf{B} = \mathbf{I}) \\
&= \mathrm{tr}\left(\mathbf{W}^\mathsf{T}\mathbf{W}\right) - \mathrm{tr}\left(\mathbf{B}^\mathsf{T}\mathbf{WW}^\mathsf{T}\mathbf{B}\right).
\end{aligned} \tag{13}$$

Based on this we can recover the optimum basis matrix $\mathbf{B}$ as the maximizer of the constrained problem:

$$\mathbf{B}^\star = \arg\max_{\mathbf{B}} \mathrm{tr}\left(\mathbf{B}^\mathsf{T}\mathbf{WW}^\mathsf{T}\mathbf{B}\right) \ \text{ s.t } \mathbf{B}^\mathsf{T}\mathbf{B} = \mathbf{I}. \tag{14}$$

The above maximization problem enjoys a closed-form solution Fan (1949), which is fully defined by the eigenvalues of the matrix $\mathbf{P} = \mathbf{WW}^\mathsf{T}$. Specifically, the matrix $\mathbf{P} \in \mathbb{R}^{d_1 \times d_1}$, which is symmetric and positive semi-definite, admits the eigenvalue decomposition $\mathbf{P} = \mathbf{U\Lambda U}^\mathsf{T}$, with $\mathbf{U} \in \mathbb{R}^{d_1 \times d_1}$ holding the eigenvectors of $\mathbf{P}$ in its columns. Then, the maximizer of Eq. (14) is recovered as $\mathbf{B}^\star = \mathbf{U}_r$ where $\mathbf{U}_r \in \mathbb{R}^{d_1 \times r}$ is a reduced version of $\mathbf{U}$ formed with the $r$ eigenvectors corresponding to the largest eigenvalues of $\mathbf{P}$. A useful observation is that the eigenvectors of $\mathbf{P}$ exactly match the left-singular vectors of $\mathbf{W} \in \mathbb{R}^{d_1 \times d_2}$. Indeed, if $\mathbf{W}$ admits the singular value decomposition $\mathbf{W} = \mathbf{U\Sigma V}^\mathsf{T}$, then we have that: $\mathbf{P} = \mathbf{WW}^\mathsf{T} = \mathbf{U\Sigma}^2\mathbf{U}^\mathsf{T} \equiv \mathbf{U\Lambda U}^\mathsf{T}$, with $\mathbf{\Lambda} = \mathbf{\Sigma}^2$. Therefore, instead of performing the eigenvalue decomposition on $\mathbf{P}$ we can recover $\mathbf{U}$ and consequently $\mathbf{U}_r$ by computing the SVD of $\mathbf{W}$.

Finally, we can compute the optimum coefficient matrix as:

$$\begin{aligned}
\mathbf{C}^\star &= \left(\mathbf{B}^\star\right)^\mathsf{T}\mathbf{W} = \mathbf{U}_r^\mathsf{T}\mathbf{W} \\
&= \mathbf{U}_r^\mathsf{T}\overbrace{\begin{bmatrix}\mathbf{U}_r & \mathbf{U}_{d-r}\end{bmatrix}}^{\mathbf{U}}\overbrace{\begin{bmatrix}\mathbf{\Sigma}_r & \mathbf{O}_{r\times(d-r)} \\ \mathbf{O}_{(d-r)\times r} & \mathbf{\Sigma}_{d-r}\end{bmatrix}}^{\mathbf{\Sigma}}\overbrace{\begin{bmatrix}\mathbf{V}_r^\mathsf{T} \\ \mathbf{V}_{d-r}^\mathsf{T}\end{bmatrix}}^{\mathbf{V}^\mathsf{T}} \\
&= \begin{bmatrix}\mathbf{I}_{r\times r} & \mathbf{O}_{r\times(d-r)}\end{bmatrix}\begin{bmatrix}\mathbf{\Sigma}_r\mathbf{V}_r^\mathsf{T} \\ \mathbf{\Sigma}_{d-r}\mathbf{V}_{d-r}^\mathsf{T}\end{bmatrix} \\
&= \mathbf{\Sigma}_r\mathbf{V}_r^\mathsf{T},
\end{aligned} \tag{15}$$

where $\mathbf{V}_r \in \mathbb{R}^{d_2 \times r}$ is a reduced version of $\mathbf{V}$ formed with the $r$ right singular vectors of $\mathbf{W}$ that correspond to its top-$r$ singular values, which are kept in the diagonal matrix $\mathbf{\Sigma}_r \in \mathbb{R}^{r \times r}$.

## A.2    DATA-AWARE LOW-RANK WEIGHT APPROXIMATION

While the low-rank approximation of the weights $\mathbf{W}$ has been extensively used for compression tasks, in practice is not well suited to LLMs and it can lead to a severe drop of their performance . Several recent works have suggested that instead of approximating the weights $\mathbf{W}$ with a low-rank matrix, a more efficient strategy is to model the weight activations, $\mathbf{Z} = \mathbf{XW}$, as low-rank. Here, the matrix $\mathbf{X} = \begin{bmatrix}\mathbf{x}_1 & \dots & \mathbf{x}_N\end{bmatrix}^\mathsf{T} \in \mathbb{R}^{N \times d}$ holds in its rows the $d$-dimensional input vectors $\mathbf{x}_n$

with $n = 1 \ldots, N$, which play the role of calibration data. Under this modeling framework, we can approximate the matrix weights $\mathbf{W}$ as the minimizer of the following problem:

$$\tilde{\mathbf{W}}^* = \arg\min_{\tilde{\mathbf{W}}} \left\| \mathbf{XW} - \mathbf{X\tilde{W}} \right\|_F \text{ s.t. } \text{rank}\left( \mathbf{X\tilde{W}} \right) = r. \tag{16}$$

Let us now consider $\mathbf{Y} = \mathbf{XL}^{-1} \in \mathbb{R}^{N \times d_1}$ to be a semi-orthogonal matrix (column-orthogonal matrix), that is $\mathbf{Y}^\top \mathbf{Y} = \mathbf{I}_{d_1}$, which is obtained by linearly transforming the matrix $\mathbf{X}$ using a non-singular matrix $\mathbf{L} \in \mathbb{R}^{d_1 \times d_1}$. Here we assume that $N \geq d$ and the matrix $\mathbf{X}$ is of full rank. We note that there are different ways we can achieve this column-orthogonalization of $\mathbf{X}$. Among them we can employ the QR/SVD decomposition on $\mathbf{X}$ and the Cholesky/Eigen-value decomposition on $\mathbf{X}^\top \mathbf{X}$ to compute a proper linear transformation $\mathbf{L}$. By using the representation $\mathbf{X} = \mathbf{YL}$ we can rewrite the problem of Eq. (16) as:

$$\tilde{\mathbf{W}}^* = \arg\min_{\tilde{\mathbf{W}}} \left\| \mathbf{YLW} - \mathbf{YL\tilde{W}} \right\|_F \text{ s.t. } \text{rank}\left( \mathbf{YL\tilde{W}} \right) = r. \tag{17}$$

To solve the above minimization problem we first note that due to the orthonormal columns of $\mathbf{Y}$, it can be expressed in the equivalent form:

$$\tilde{\mathbf{W}}^* = \arg\min_{\tilde{\mathbf{W}}} \left\| \boldsymbol{\Delta} - \mathbf{L\tilde{W}} \right\|_F \text{ s.t. } \text{rank}\left( \mathbf{L\tilde{W}} \right) = r, \tag{18}$$

where $\boldsymbol{\Delta} = \mathbf{LW}$. Next, we introduce the auxiliary matrix $\tilde{\boldsymbol{\Delta}} = \mathbf{L\tilde{W}}$ and the problem in Eq. (18) becomes:

$$\tilde{\boldsymbol{\Delta}}^* = \arg\min_{\tilde{\boldsymbol{\Delta}}} \left\| \boldsymbol{\Delta} - \tilde{\boldsymbol{\Delta}} \right\|_F \text{ s.t. } \text{rank}\left( \tilde{\boldsymbol{\Delta}} \right) = r, \tag{19}$$

which is the orthogonal projection of $\boldsymbol{\Delta}$ to the space of $r$-rank matrices. Given that $\tilde{\boldsymbol{\Delta}}^* = \mathbf{L\tilde{W}}^*$ and $\mathbf{L}$ is invertible, we can now recover $\tilde{\mathbf{W}}^* = \mathbf{L}^{-1} \tilde{\boldsymbol{\Delta}}^*$.

To conclude, if $\boldsymbol{\Delta}$ admits the singular value decomposition $\boldsymbol{\Delta} = \mathbf{U\Sigma V}^\top$, then the optimal $r$-rank approximation of $\mathbf{W}$ that minimizes the loss in Eq. (16) can be written in the form:

$$\tilde{\mathbf{W}} = \mathbf{BC} = \underbrace{\mathbf{L}^{-1}\mathbf{U}_r}_{\mathbf{B}} \underbrace{\boldsymbol{\Sigma}_r \mathbf{V}_r^\top}_{\mathbf{C}}. \tag{20}$$

We note that in this case, unlike the direct weight low-rank approximation, the matrix $\mathbf{B} = \mathbf{L}^{-1}\mathbf{U}_r$ does not correspond to a basis of a subspace of $\mathbb{R}^{d_1}$, since its columns are no longer orthonormal, that is $\mathbf{B}^\top \mathbf{B} = \mathbf{U}_r^\top \left( \mathbf{LL}^\top \right)^{-1} \mathbf{U}_r \neq \mathbf{I}$.

### A.3 PSEUDO ALGORITHM OF THE PROPOSED METHOD

**Goal.** Given a weight matrix $\mathbf{W} \in \mathbb{R}^{d_1 \times d_2}$ and a small calibration set $\mathbf{X} \in \mathbb{R}^{N \times d_1}$, compute an activation-aware sparse–dictionary factorization $\mathbf{W} \approx \widetilde{\mathbf{W}} = \mathbf{DS}$ under a target compression ratio. The procedure consists of whitening the activation objective, alternating sparse coding and dictionary updates on the whitened weights, and a final de-whitening step.

**(1) Calibration and whitening.** Compute an invertible transform $\mathbf{L} \in \mathbb{R}^{d_1 \times d_1}$ (e.g., via QR/SVD of $\mathbf{X}$ or Cholesky of $\mathbf{X}^\top \mathbf{X}$) such that $\mathbf{Y} = \mathbf{XL}^{-1}$ has orthonormal columns ($\mathbf{Y}^\top \mathbf{Y} = \mathbf{I}$). Left-multiply $\mathbf{W}$ to obtain the whitened weights $\mathbf{W}_L = \mathbf{LW}$. Whitening converts the data-aware loss $\|\mathbf{XW} - \mathbf{X}\widehat{\mathbf{W}}\|_F^2$ into a standard Frobenius objective $\|\mathbf{W}_L - \mathbf{D}_L \mathbf{S}\|_F^2$ that is amenable to dictionary learning.

**(2) Initialization.** Initialize the whitened dictionary $\mathbf{D}_\mathbf{L}^{(0)} \in \mathbb{R}^{d_1 \times k}$ (e.g., random permutation of columns of $\mathbf{W}$) and set $\mathbf{S}^{(0)} = 0$. The pair $(k, s)$ is set from the target compression ratio via Eq. 10 optionally using the fixed ratio $\rho = k/s$.

**(3) Alternating minimization.** Repeat for $t = 1, \ldots, T$: *(a) Sparse coding.* For each column $j$, solve

$$\mathbf{s_j^{(t)}} \in \arg \min_{\|\mathbf{s}\|_0 \leq s} \left\| (\mathbf{W}_L)_{:,j} - \mathbf{D_L}^{(t-1)}\mathbf{s} \right\|_2^2,$$

using OMP (greedy selection with orthogonal residual updates) to enforce exactly $s$ nonzeros per column. *(b) Dictionary update.* For each atom $i$, collect its support $\Omega_i = \{j : s_{i,j}^{(t)} \neq 0\}$ and form the residual on those columns:

$$\mathbf{R}_i = \mathbf{W}_L[:, \Omega_i] - \sum_{\ell \neq i} \mathbf{D_{L,\ell}}^{(t-1)} \mathbf{s}_{\ell,\Omega_i}^{(t)}.$$

Update $(\mathbf{D_{L,i}}^{(t)}, \mathbf{s}_{i,\Omega_i}^{(t)})$ by the best rank-1 approximation $\mathbf{R}_i \approx \mathbf{u}\,\sigma\,\mathbf{v}^\top$ (set $/mD_{L,i}^{(t)} \leftarrow \mathbf{u}$, $\mathbf{s}_{i,\Omega_i}^{(t)} \leftarrow \sigma\mathbf{v}^\top$). This preserves the current sparsity pattern while reducing the residual. Iterate atoms sequentially. Stop when the maximum iteration $T$ is reached or when the relative improvement falls below a tolerance.

**(4) De-whitening and packing.** Map the dictionary back to activation space via $\mathbf{D_a} = \mathbf{L}^{-1}\mathbf{D_L^{(T)}}$ and set $\widetilde{\mathbf{W}} = \mathbf{D}_a\mathbf{S}^{(T)}$. For storage, keep $\mathbf{D}_a$ and the $sd_2$ nonzero entries of $\mathbf{S}$ in `bf16` along with a packed binary mask $\mathbf{M} \in \{0,1\}^{k \times d_2}$ for locations (one bit per entry; $\frac{kd_2}{16}$ words). This yields the compression ratio in Appendix A.4 and Eq. 9.

**(5) Inference.** At runtime, apply $\widetilde{\mathbf{W}}$ as $\mathrm{matmul}(\mathbf{x}, \mathbf{D}_a\mathbf{S})$ with sparse–dense kernels. Reuse inner products $\langle \mathbf{x}, \mathbf{D_{a,:,i}} \rangle$ across columns to achieve the complexity in Appendix A.5; the number of active atoms controls the practical speedup.

**(6) Cross-layer variant.** For a group of layers $\mathbf{G}$, concatenate weights horizontally $\mathbf{W_G} = [\mathbf{W}_{\ell_1} \cdots \mathbf{W}_{\ell_L}]$ and stack calibration batches vertically to form $\mathbf{X}_G$. Compute $\mathbf{L}$ from $\mathbf{X}_G$, run the same alternating procedure on $\mathbf{W}_{L,G} = \mathbf{L}\mathbf{W}_G$ to obtain a single shared $\mathbf{D}_a$, and slice the corresponding blocks of $\mathbf{S}_G$ back to per-layer coefficients.

## A.4 DERIVATION OF THE COSPADI COMPRESSION RATIO

We derive the expression for the compression ratio $\gamma^{\mathrm{SD}}$ of our sparse–dictionary (SD) parameterization. Let $\mathbf{W} \in \mathbb{R}^{d_1 \times d_2}$ be factorized as

$$\mathbf{W} \approx \mathbf{DS}, \qquad \mathbf{D} \in \mathbb{R}^{d_1 \times k}, \ \ \mathbf{S} \in \mathbb{R}^{k \times d_2},$$

where each column of $\mathbf{S}$ has exactly $s$ nonzeros (*column-s-sparse*). Throughout, we store real values in `bfloat16` (16 bits) as is common in modern LLMs.

**Dense baseline.** A dense $\mathbf{W}$ requires $d_1 d_2$ `bf16` values.

**Dictionary term.** The dictionary $\mathbf{D}$ stores $d_1 k$ `bf16` values.

**Sparse codes.** Naively, $\mathbf{S}$ would need $kd_2$ values. Since $\mathbf{S}$ is column-$s$-sparse, only $sd_2$ values are stored. For locations, one option is COO: per nonzero we keep a row index and (redundantly) the column index. Because sparsity is fixed per column, column indices can be omitted; keeping only row indices yields $sd_2$ indices. With 16-bit indices, the total becomes $sd_2$ values $+ sd_2$ indices $= 2sd_2$ 16-bit words. For typical $\rho := k/s = 2$, this equals $kd_2$ words—offering no savings over dense $\mathbf{S}$ storage.

Instead, we use a *bit mask* $\mathbf{M} \in \{0,1\}^{k \times d_2}$ to mark nonzero positions. This requires $kd_2$ bits, i.e., $kd_2/16$ 16-bit words after packing. We then store $sd_2$ `bf16` values for the nonzeros and the packed mask for their positions.

**Total and ratio.** The SD parameterization thus stores

$$\underbrace{d_1 k}_{\text{dictionary}} + \underbrace{sd_2}_{\text{values}} + \underbrace{\frac{kd_2}{16}}_{\text{mask}}$$

**Algorithm 1:** Pseudo algorithm of the proposed CoSpaDi which consists of two steps: (a) sparse coding to compute the coefficients and (b) sequential dictionary update step.

**Input** : $\mathbf{W} \in \mathbb{R}^{d_1 \times d_2}$: weight matrix to compress
$\quad\quad\quad \mathbf{X} \in \mathbb{R}^{N \times d_1}$: calibration input data ($N$ samples)
$\quad\quad\quad k$: dictionary size (number of atoms, $k \geq s$)
$\quad\quad\quad s$: sparsity level (max nonzeros per column in $\mathbf{S}$)
$\quad\quad\quad T$: number of K-SVD iterations
**Output:** $\mathbf{D}_a \in \mathbb{R}^{d_1 \times k}$: activation-aware dictionary
$\quad\quad\quad \mathbf{S} \in \mathbb{R}^{k \times d_2}$: sparse coefficient matrix
$\quad\quad\quad \tilde{\mathbf{W}} = \mathbf{D}_a \mathbf{S}$: compressed weight matrix
Compute $\mathbf{L} \in \mathbb{R}^{d_1 \times d_1}$ such that $\mathbf{Y} = \mathbf{X}\mathbf{L}^{-1}$ satisfies $\mathbf{Y}^\mathsf{T}\mathbf{Y} = \mathbf{I}_{d_1}$;
`% e.g., via QR:` $\mathbf{X} = \mathbf{QR} \Rightarrow \mathbf{L} = \mathbf{R}$
`% e.g., via Cholesky:` $\mathbf{X}^\mathsf{T}\mathbf{X} = \mathbf{C}^\mathsf{T}\mathbf{C} \Rightarrow \mathbf{L} = \mathbf{C}$
$\mathbf{W}_L \leftarrow \mathbf{L}\mathbf{W}$;
Initialize $\mathbf{D}_L^{(0)} \in \mathbb{R}^{d_1 \times k}$ with random Gaussian or SVD-based atoms;
Initialize $\mathbf{S}^{(0)} \in \mathbb{R}^{k \times d_2}$ as zero matrix;
**for** $t = 1$ **to** $T$ **do**
$\quad$ **for** $j = 1$ **to** $d_2$ **do**
$\quad\quad \mathbf{s}_j^{(t)} \leftarrow \arg\min_{\|\mathbf{s}\|_0 \leq s} \left\| \mathbf{W}_{L,j} - \mathbf{D}_L^{(t-1)}\mathbf{s} \right\|_2^2$;
$\quad\quad$ `% Solve via OMP, LASSO, or thresholding`
$\quad$ **end**
$\quad$ **for** $i = 1$ **to** $k$ **do**
$\quad\quad \Omega_i \leftarrow \left\{ j \mid s_{i,j}^{(t)} \neq 0 \right\}$;
$\quad\quad$ **if** $\Omega_i \neq \emptyset$ **then**
$\quad\quad\quad \mathbf{R}_i \leftarrow \mathbf{W}_L[:,\Omega_i] - \sum_{l \neq i} \mathbf{d}_{L,l}^{(t-1)} \mathbf{s}_{l,\Omega_i}^{(t)}$;
$\quad\quad\quad [\mathbf{u}, \sigma, \mathbf{v}] \leftarrow$ rank-1 SVD of $\mathbf{R}_i$;
$\quad\quad\quad \mathbf{d}_{L,i}^{(t)} \leftarrow \mathbf{u}$;
$\quad\quad\quad \mathbf{s}_{i,\Omega_i}^{(t)} \leftarrow \sigma \cdot \mathbf{v}^\mathsf{T}$;
$\quad\quad$ **end**
$\quad$ **end**
**end**
$\mathbf{D}_a \leftarrow \mathbf{L}^{-1}\mathbf{D}_L^{(T)}$;
$\tilde{\mathbf{W}} \leftarrow \mathbf{D}_a \mathbf{S}^{(T)}$;
**return** $\mathbf{D}_a, \mathbf{S}^{(T)}, \tilde{\mathbf{W}}$;

16-bit words. Relative to the dense baseline $d_1 d_2$, the resulting compression ratio is

$$\gamma^{\mathrm{SD}} := 1 - \frac{\overbrace{d_1 k}^{\text{dict. (bf16)}} + \overbrace{s d_2}^{\text{codes (bf16)}} + \overbrace{(k d_2)/16}^{\text{mask (1 bit/entry)}}}{d_1 d_2}.$$

This matches the expression used in the main text and makes explicit the dependence on the two design knobs $(k, s)$.

## A.5 LOW-RANK AND COSPADI INFERENCE COMPLEXITY

Here we derive the multiplication complexity for the original weight, SVD-compressed weight, and dictionary-learning (k-SVD) compression. We count multiplications only (additions are of the same order). Let $\mathbf{W} \in \mathbb{R}^{d_1 \times d_2}$ be a projection matrix in some layer and $\mathbf{X} \in \mathbb{R}^{N \times d_1}$ be an input feature map; then a dense product $\mathbf{Y} = \mathbf{X}\mathbf{W}$ costs

$$O_{baseline} = N d_1 d_2. \tag{21}$$

For low-rank, in particular, SVD compression with rank $r$ the projection matrix is approximated with two matrices $\mathbf{W} \approx \mathbf{U}\mathbf{V}$ with $\mathbf{U} \in \mathbb{R}^{d_1 \times r}$ and $\mathbf{V} \in \mathbb{R}^{r \times d_2}$, resulting in the following complexity:

$$O_{LR} = Nd_1 r + Nr d_2 = Nr(d_1 + d_2). \tag{22}$$

Sparse dictionary (SD) learning similarly represents $\mathbf{W} \approx \mathbf{DS}$ with dictionary $\mathbf{D} \in \mathbb{R}^{d_1 \times k}$ of $k$ atoms and sparse coefficient matrix $\mathbf{S} \in \mathbb{R}^{k \times d_2}$. Omitting sparsity of $\mathbf{S}$ will result in:

$$O_{SD,dense} = Nd_1 k + Nk d_2 = Nk(d_1 + d_2). \tag{23}$$

Taking into account that each column $\mathbf{s}_j$ of $\mathbf{S}$ is $s$-sparse, the $(i,j)$ element of $\mathbf{Y} = \mathbf{XDS}$ is

$$y_{i,j} = \sum_{k=1}^{K} \mathbf{S}_{k,j} \langle \mathbf{X}_{i,:}, \mathbf{D}_{:,k} \rangle = \sum_{k \in \mathbf{S}_j} \mathbf{S}_{k,j} \langle \mathbf{X}_{i,:}, \mathbf{D}_{:,k} \rangle, \tag{24}$$

where $S_j = \text{supp}(s_j)$ and $|\mathbf{S}_j| = s$. The overall sparse complexity depends on whether the inner products $\langle \mathbf{X}_{i,:}, \mathbf{D}_{:,k} \rangle$ are reused across columns. With the most efficient way with reuse letting $\mathbf{U} = \bigcup_{j=1}^{d_2} S_j$ and $K_{active} = |U|$ we have:

$$O_{SD,sparse-reuse} = Nd_1 K_{active} + Nsd_2, \qquad s \leq K_{active} \leq \min(K, sd_2). \tag{25}$$

With proposed truncation of 2-bits in mantissa we can omit term for storing indices of nonzero elements in $\mathbf{S}$ resulting in corrected compression ratio:

$$\hat{\gamma}^{SD} = 1 - \frac{d_1 k + sd_2}{d_1 d_2}$$

The rank for the low-rank decomposition could be estimated from the compression ratio with the following equation:

$$r = \frac{(1 - \gamma^{LR})d_1 d_2}{d_1 + d_2}$$

For sparse dictionary based method we defined $\rho = k/s$ and, thus we can estimate both $k$ and $s$ in the following way:

$$k = \frac{(1 - \hat{\gamma}^{SD}) d_1 d_2}{d_1 + \frac{d_2}{\rho}}, \qquad s = \frac{k}{\rho}.$$

Under the same compression ratio for both SVD and CoSpaDi we obtain exactly the same complexity:

$$O_{LR} = Nr(d_1 + d_2) = N(1 - \gamma^{LR})d_1 d_2$$

$$O_{SD} = Nd_1 k + Nsd_2 = N(d_1 k + d_2 \frac{k}{\rho}) = Nk(d_1 + \frac{d_2}{\rho}) = N\frac{(1 - \hat{\gamma}^{SD}) d_1 d_2}{d_1 + \frac{d_2}{\rho}}(d_1 + \frac{d_2}{\rho}) = N(1 - \hat{\gamma}^{SD}) d_1 d_2$$

$$O_{LR} = O_{SD} = N(1 - \gamma)d_1 d_2. \tag{26}$$

While, theoretical complexity is the same, in practice inference time depends on the sparsity structure ($K_{\text{active}}$), indexing overhead, and kernel efficiency, as can be observed from Figs. 4, 5 and 6.

### A.6 COMPARISON WITH OTHER TRAINING-FREE METHODS

We also evaluate the performance of the proposed CoSpaDi relative to other training free methods, particularly structural pruning ones and ASVD Yuan et al. (2023) which is another data-aware SVD-based method. The results are provided in Table 5.

Across LLaMA-3 8B, our method consistently outperforms structural pruning baselines at moderate budgets. At CR $\approx 0.2$, CoSpaDi attains the best average accuracy (0.618 vs. 0.580 for ReplaceMe and 0.551 for LLM-Pruner) while also delivering the lowest perplexities, indicating balanced generative and discriminative quality. At CR $\approx 0.3$, CoSpaDi widens the margin in average accuracy (0.537 vs. 0.469/0.405) and avoids the severe perplexity blow-ups observed for pruning methods. Even under aggressive compression (CR $\approx 0.4$–$0.5$), CoSpaDi remains competitive or superior on most tasks and preserves far more stable perplexity profiles, whereas ReplaceMe and LLM-Pruner exhibit task collapse (near-zero LAMBADA and extreme PPL spikes). Overall, the results support dictionary–sparsity as a more robust data-free alternative to structural pruning at matched or tighter budgets, especially when both accuracy and perplexity must be maintained.

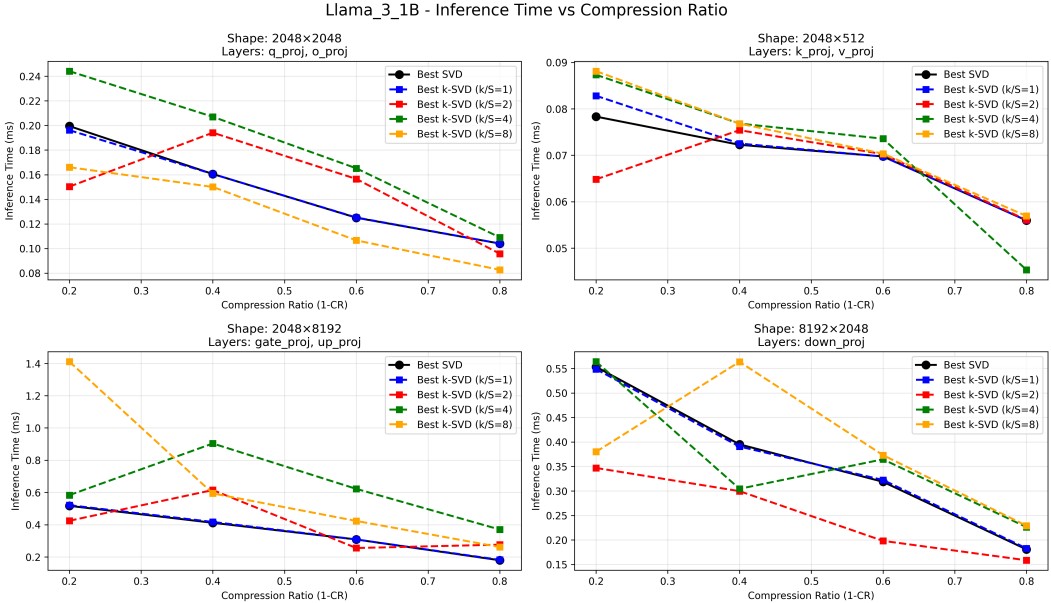

Figure 4: Inference time for different projection layers of Llama3.2 1B for different compression ratios and $k/s$ ratios on A100

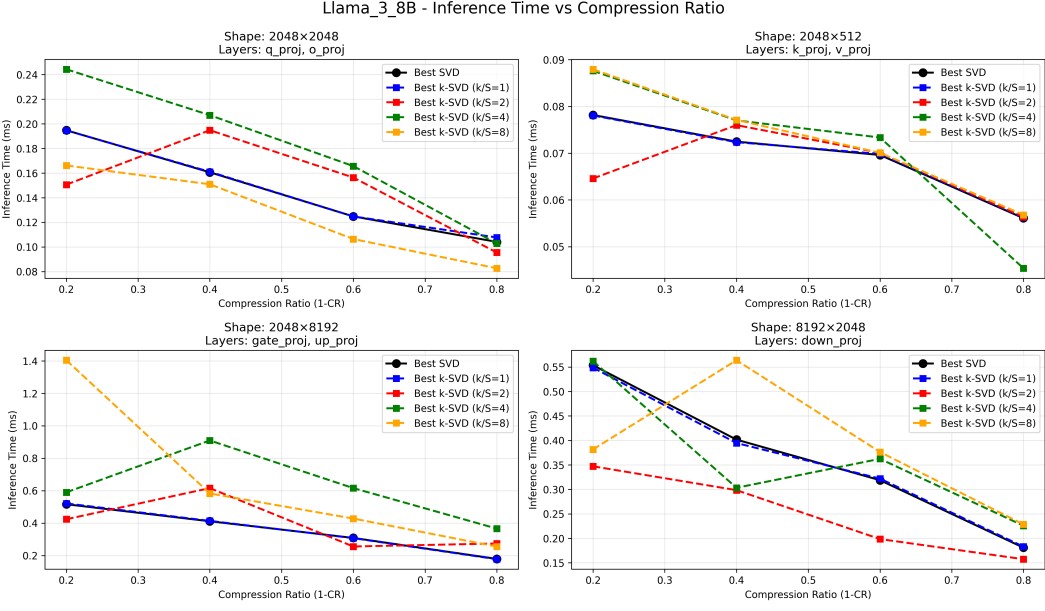

Figure 5: Inference time for different projection layers of Llama3 8B for different compression ratios and $k/s$ ratios on A100

## A.7 CROSS-LAYER SCENARIO

In Table 6, we report the dictionary size ($k$), sparsity level ($s$), and input/output weight dimensions for each linear transform under a fixed $k/s$ ratio. This includes the query, key, value, and output projections within the attention block, as well as the up, down, and gated projections in the gated MLP block. The dictionary and sparsity dimensions are adjusted according to the target compression rate.

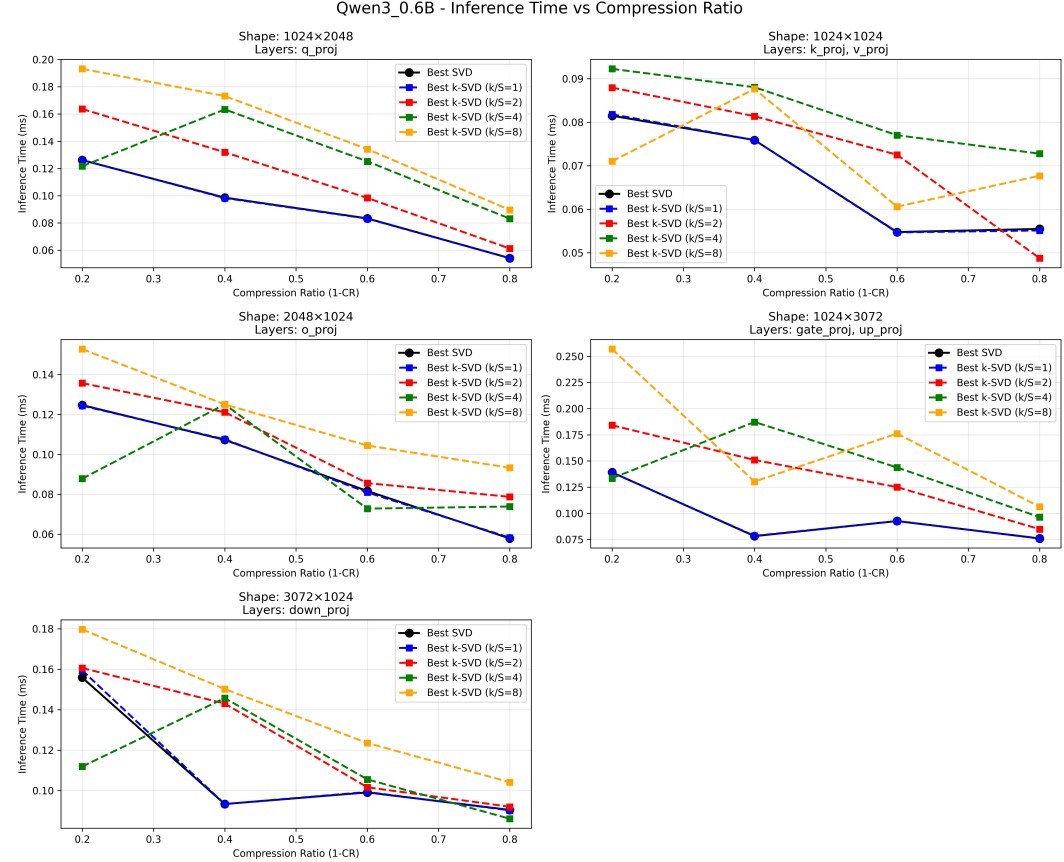

Figure 6: Inference time for different projection layers of Qwen3 0.6B for different compression ratios and $k/s$ ratios on A100

## A.8 K-SVD AND POWER ITERATION ANALYSIS

We conducted ablation studies to assess the effect of the number of K-SVD iterations and power iterations on performance using Llama3.2-1B with fixed $\rho = 2$. The left plot in Fig. 7 shows that average accuracy stabilizes after roughly 50 K-SVD iterations, while perplexity continues to decrease slightly before flattening out. The right plot of Fig. 7 indicates that very few power iterations are sufficient for stable convergence: performance improves sharply up to around 5 iterations, after which additional iterations yield minimal benefit. Based on these results, we fixed the number of K-SVD iterations to 60 and power iterations to 8 in our final experiments, which provides a good balance between accuracy, perplexity, and computational efficiency.

## A.9 USE OF LARGE LANGUAGE MODELS

The authors acknowledge the use of a large language model (LLM) solely for language editing and grammatical refinement of the current manuscript. All scientific content, analysis, and interpretations presented herein are the sole responsibility of the authors.

Table 5: Comparison of the proposed CoSpaDi method with other training-free methods including ASVD Yuan et al. (2023) and state-of-the-art structured pruning methods ReplaceMe Shopkhoev et al. (2025) and LLM-Pruner Ma et al. (2023) on Llama3 8B under different compression ratios. We report accuracy on different benchmarks as well as its average and perplexity. Best results are highlighted with **bold**

| Method | CR | Accuracy↑ | | | | | | | | | Perplexity↓ | |
|---|---|---|---|---|---|---|---|---|---|---|---|---|
| | | PIQA | Hella Swag | LAMBADA | ARC-e | ARC-c | SciQ | Race | MMLU | Avg. | Wiki Text | LAMBADA |
| **Llama3 8B** | | 80.7 | 79.1 | 75.6 | 77.7 | 53.5 | 93.9 | 40.3 | 62.2 | 70.4 | 7.3E+00 | 3.1E+00 |
| ASVD | 0.2 | 55.8 | 26.7 | 0.7 | 32.4 | 21.5 | 30.5 | 21.8 | 23.5 | 26.6 | 9.4E+04 | 9.9E+05 |
| ReplaceMe | 0.22 | 73.1 | 65.7 | 42.1 | 65.9 | **43.7** | 86.4 | 35.4 | **51.7** | 58.0 | 3.4E+01 | 2.0E+01 |
| LLM-Pruner | 0.2 | **75.5** | **67.5** | 51.0 | 62.1 | 36.6 | 87.8 | 35.1 | 25.0 | 55.1 | **1.6E+01** | 1.1E+01 |
| CoSpaDi | | 75.2 | 66.5 | **73.8** | 66.5 | 41.6 | 89.5 | 38.2 | 42.8 | **61.8** | 2.0E+01 | **4.3E+00** |
| ASVD | 0.3 | 53.6 | 26.0 | 0.0 | 26.9 | 21.4 | 21.9 | 21.2 | 23.5 | 24.3 | 3.4E+05 | 1.4E+07 |
| ReplaceMe | 0.31 | 66.6 | 53.8 | 24.0 | 50.7 | **37.9** | 77.3 | 34.0 | 30.6 | 46.9 | 6.7E+01 | 1.3E+02 |
| LLM-Pruner | 0.3 | 67.3 | 45.1 | 20.9 | 45.4 | 28.8 | 63.4 | 30.1 | 22.9 | 40.5 | **3.8E+01** | 2.2E+02 |
| CoSpaDi | | **70.5** | **56.2** | **61.3** | **54.2** | 33.5 | **85.7** | 36.2 | 32.2 | **53.7** | 4.5E+01 | **9.2E+00** |
| ASVD | 0.4 | 53.7 | 25.8 | 0.0 | 25.0 | 22.9 | 20.8 | 22.0 | 24.3 | 24.3 | 2.1E+05 | 2.6E+07 |
| ReplaceMe | 0.41 | 61.7 | **44.3** | 9.8 | 37.4 | **27.5** | 60.4 | **31.6** | 26.4 | 37.4 | 2.3E+02 | 1.8E+03 |
| LLM-Pruner | 0.4 | 50.3 | 25.8 | 1.5 | 26.4 | 25.8 | 28.1 | 21.8 | 23.2 | 25.4 | ∞ | 5.7E+05 |
| CoSpaDi | | **63.7** | 41.4 | **30.3** | **39.1** | 26.6 | **68.5** | 30.5 | 25.4 | **40.7** | **1.8E+02** | **1.2E+02** |
| ASVD | 0.5 | 52.0 | 25.5 | 0.0 | 26.0 | 22.6 | 19.8 | 21.3 | **25.5** | 24.1 | 1.7E+06 | 1.7E+07 |
| ReplaceMe | | **57.2** | **36.4** | 3.7 | 29.8 | **25.4** | **38.1** | **27.9** | 23.0 | **30.2** | 7.0E+02 | 5.0E+04 |
| LLM-Pruner | | 50.4 | 26.2 | 0.4 | 26.3 | 26.4 | 25.6 | 21.7 | 23.7 | 25.1 | 1.2E+03 | 1.2E+06 |
| CoSpaDi | | 56.4 | 31.0 | **3.9** | 29.9 | 22.2 | 37.7 | 22.9 | 23.0 | 28.4 | **8.1E+02** | **7.6E+03** |

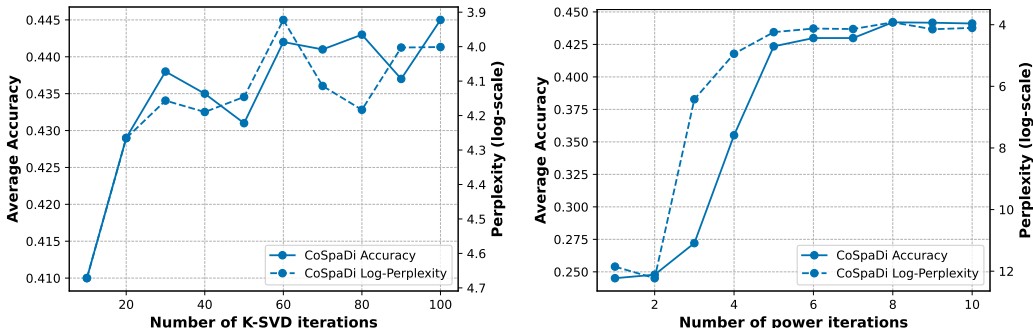

Figure 7: Average benchmark accuracy and WikiText perplexity with respect to the number of K-SVD iterations (left) and the number of power iterations (right) for Llama3.2-1B with $\rho = 2$

Table 6: Compression configurations for Llama2 7B weight matrices across varying compression rates (20%–50%). Each row specifies sparsity parameters ($k$, $s$, $k/s$ ratio) and weight dimensions ($d_1$, $d_2$) for different weight types (Query, Key, Value, Up, Gate, Down, Out), grouped by compression rate and group size (1 or 2).

| Weight Type | Compression Rate | Group size | Dictionary, $k$ | Sparsity, $s$ | $k/s$ ratio | $d_1$ | $d_2$ |
|---|---|---|---|---|---|---|---|
| Query | | | 3276 | 1638 | 2 | 4096 | 8192 |
| Key | | | 3276 | 1638 | 2 | 4096 | 8192 |
| Value | | 2 | 3276 | 1638 | 2 | 4096 | 8192 |
| Up | 20% | | 4776 | 2388 | 2 | 4096 | 22016 |
| Gate | | | 4776 | 2388 | 2 | 4096 | 22016 |
| Down | | 1 | 2762 | 1381 | 2 | 11008 | 4096 |
| Out | | | 2184 | 1092 | 2 | 4096 | 4096 |
| Query | | | 2866 | 1433 | 2 | 4096 | 8192 |
| Key | | | 2866 | 1433 | 2 | 4096 | 8192 |
| Value | | 2 | 2866 | 1433 | 2 | 4096 | 8192 |
| Up | 30% | | 4178 | 2089 | 2 | 4096 | 22016 |
| Gate | | | 4178 | 2089 | 2 | 4096 | 22016 |
| Down | | 1 | 2416 | 1208 | 2 | 11008 | 4096 |
| Out | | | 1910 | 955 | 2 | 4096 | 4096 |
| Query | | | 2456 | 1228 | 2 | 4096 | 8192 |
| Key | | | 2456 | 1228 | 2 | 4096 | 8192 |
| Value | | 2 | 2456 | 1228 | 2 | 4096 | 8192 |
| Up | 40% | | 3582 | 1791 | 2 | 4096 | 22016 |
| Gate | | | 3582 | 1791 | 2 | 4096 | 22016 |
| Down | | 1 | 2072 | 1036 | 2 | 11008 | 4096 |
| Out | | | 1638 | 819 | 2 | 4096 | 4096 |
| Query | | | 2048 | 1024 | 2 | 4096 | 8192 |
| Key | | | 2048 | 1024 | 2 | 4096 | 8192 |
| Value | | 2 | 2048 | 1024 | 2 | 4096 | 8192 |
| Up | 50% | | 2984 | 1492 | 2 | 4096 | 22016 |
| Gate | | | 2984 | 1492 | 2 | 4096 | 22016 |
| Down | | 1 | 1726 | 863 | 2 | 11008 | 4096 |
| Out | | | 1364 | 682 | 2 | 4096 | 4096 |

