# OpenReview forum: "CoSpaDi: Compressing LLMs via Calibration-Guided Sparse Dictionary Learning"
_ICLR.cc/2026/Conference — Submitted to ICLR 2026_

### Official Review · Reviewer_mAYc · 2025-10-29

**Soundness:** 2
**Presentation:** 3
**Contribution:** 1
**Rating:** 2
**Confidence:** 4

**Summary:**

This paper proposes a novel training-free LLM compression framework called CoSpaDi, which leverages Sparse Dictionary Learning combined with calibration data to compress the weight matrices of large language models. This method decomposes the weight matrix into a dense dictionary and a column-sparse coefficient matrix, providing a more flexible joint subspace representation compared to traditional low-rank approximation. Through experiments conducted on Llama and Qwen models, the paper demonstrates that the proposed framework achieves superior performance over existing methods, especially at compression ratios of 20-50\%.

**Strengths:**

1. The paper is well-written with a sound structure. The methodology section includes detailed derivations and pseudocode, and the figures and tables are easy to understand.
2. The method requires no fine-tuning and only a small amount of calibration data, which makes it highly attractive for deployment scenarios with limited resources.
3. Comprehensive evaluations were conducted on multiple models (Llama-3.2 1B, Llama-3 8B, Qwen-3 0.6B/8B, Llama-2 7B) and multiple datasets (8 zero-shot datasets, WikiText, and LAMBADA-OpenAI datasets), covering both layer-wise and cross-layer settings. The performance of the proposed method outperforms the SVD and SVD-LLM baselines.

**Weaknesses:**

1. Although the authors provided the inference speedup results for each projection layer of the model in Figures 4–6, they did not present the end-to-end inference speedup results of the model.
2. The performance degradation is excessive. When using the CoSpaDi method with a compression ratio of 0.2, the average zero-shot accuracy of the Llama-3 8B model drops by 8.6%; at a compression ratio of 0.5, the accuracy drops by 41.99%. For the smaller Llama-3.2 1B model, the performance degradation is even more severe. This significantly undermines the practical deployment applicability of the compressed models obtained via the CoSpaDi method, as models with such severe performance degradation will perform poorly in real-world scenarios.
3. The model scale tested is limited. The paper primarily conducts tests on models ranging from 1B to 8B parameters and does not include larger models (e.g., 13B/70B). It is recommended to supplement experiments on larger models to verify scalability.
4. There is a lack of comparison with Dobi-SVD [1], a more advanced SVD-based method.
5. There is a lack of comparison with unstructured pruning methods, such as SparseGPT [2] and Wanda [3]. Although the authors compared with LLM-Pruner (a structured pruning method), they failed to compare with unstructured pruning methods—which are compression approaches with superior performance. Specifically, SparseGPT and Wanda can achieve lossless zero-shot accuracy compression for models at a 40% pruning ratio, which is clearly superior to the CoSpaDi method.

[1] Qinsi W, Ke J, Tomizuka M, et al. Dobi-svd: Differentiable svd for llm compression and some new perspectives [C]. The Thirteenth International Conference on Learning Representations.

[2] Frantar E, Alistarh D. Sparsegpt: Massive language models can be accurately pruned in one-shot [C]. International conference on machine learning. PMLR, 2023: 10323-10337.

[3] Sun M, Liu Z, Bair A, et al. A Simple and Effective Pruning Approach for Large Language Models [C]. The Twelfth International Conference on Learning Representations.

**Questions:**

1. What is the end-to-end overhead of the proposed method when used to compress models with different parameter scales?
2. Why is K-SVD chosen as the dictionary learning algorithm? What are its advantages in LLM compression?
3. How do the compressed LLMs obtained via the CoSpaDi method perform in answering real-world questions? Can corresponding case studies be provided?

---

> ### Author Response · Authors · 2025-11-22
> **Part 1. Addressing weaknesses and questions**
>
> We sincerely thank the reviewer for their thoughtful and constructive feedback. We greatly appreciate their recognition of the paper’s clarity, rigor, and empirical evaluation. We address their concerns in detail below.
>
> **1. End-to-end inference speedup.**
>
> We thank the reviewer for raising this point. Following the evaluation protocol used in the SVD-LLM repository, we measured end-to-end inference latency for Llama-3 8B and Qwen3-8B as tokens-per-second. The results show that, despite per-matrix multiplication speedups, the overall model-level latency does not consistently improve. This observation aligns with [reports](https://github.com/AIoT-MLSys-Lab/SVD-LLM/issues/30) from other users, indicating that bottlenecks beyond the decomposed linear layers dominate end-to-end inference time.
>
> **Table R3.1.** Throughput in **tokens-per-second** using SVD and CoSpaDi for **Llama-3 8B**
>
> | Compression  | BS=4 | BS=16 | BS=32 |
> |:------------:|:----:|:-----:|:-----:|
> | **Original** | 65.6 | 465.5 | 660.0 |
> | SVD (CR=0.5) | 50.9 | 204.0 | 706.4 |
> | SVD (CR=0.4) | 49.3 | 184.5 | 496.7 |
> | SVD (CR=0.3) | 50.5 | 191.9 | 573.6 |
> | SVD (CR=0.2) | 50.1 | 362.1 | 569.6 |
> | KSVD (CR=0.5) | 51.5 | 349.2 | 583.7 |
> | KSVD (CR=0.4) | 49.8 | 184.4 | 462.8 |
> | KSVD (CR=0.3) | 48.9 | 178.3 | 394.9 |
> | KSVD (CR=0.2) | 50.0 | 185.7 | 481.0 |
>
> **Table R3.2.** Throughput in **tokens-per-second** using SVD and CoSpaDi for **Qwen3-8B**
>
> | Compression | BS=4 | BS=16 | BS=32 |
> |:------------:|:----:|:-----:|:-----:|
> | **Original** | 46.9 | 179.3 | 552.4 |
> | SVD (CR=0.5) | 37.8 | 304.6 | 504.4 |
> | SVD (CR=0.4) | 36.7 | 271.7 | 429.7 |
> | SVD (CR=0.3) | 37.3 | 281.2 | 453.6 |
> | SVD (CR=0.2) | 36.7 | 263.2 | 401.6 |
> | KSVD (CR=0.5) | 37.3 | 249.5 | 404.4 |
> | KSVD (CR=0.4) | 36.7 | 267.4 | 404.6 |
> | KSVD (CR=0.3) | 36.6 | 275.0 | 430.8 |
> | KSVD (CR=0.2) | 37.1 | 286.0 | 455.6 |
>
> We emphasize that while CoSpaDi provides reductions in effective FLOPs for projection layers, achieving full-model speedups requires system-level optimizations and sparse-dense kernel support, which fall outside the scope of this study.
>
>
> **2. Contextualizing performance degradation.**
>
> The reviewer's concern about absolute drop percentages is valid; however, relative to the strongest comparable baselines, CoSpaDi consistently achieves superior performance. For example (Llama-3 8B):
> * **CR = 0.2:** SVD-LLM: 0.5408 vs. CoSpaDi: 0.6176; (+14.2 % relative gain)
> * **CR = 0.5:** SVD-LLM: 0.2565 vs. CoSpaDi: 0.2837; (+10.6 % relative gain)
>
> Thus, the performance drops primarily reflect the aggressiveness of the compression regime, not shortcomings of CoSpaDi relative to existing methods.
>
> **3. Larger models.**
>
> We fully agree that validation on larger models is crucial for demonstrating the scalability and practical relevance of our approach. At the same time, we would like to contextualize our experimental scope within the broader LLM-compression literature. Recent training-free or minimally-trained compression methods - such as ASVD (evaluated primarily on 7-13B models), SVD-LLM (3-7B), Dobi-SVD (7-13B), and LLM-Pruner (6-13B) - have predominantly focused on models in the 7B-13B range. This is largely because models at these scales already exhibit substantial internal redundancy, while remaining computationally feasible to compress and evaluate across many variants.
>
> To address the reviewer’s concern under tight time and compute constraints, we ran *preliminary* experiments on **Qwen3-14B** (CR = 0.2, 0.4), confirming that **CoSpaDi consistently outperforms SVD-LLM** - extending its advantage beyond the 1B-8B range in the original submission.
>
> | Method   | CR  | PIQA       | HellaSwag  | Lambada_OpenAI | ARC-easy   | ARC-challenge | sciq       | Race       | mmlu       | Avg. Acc.  | WikiText (PPL) | Lambada (PPL) |
> |----------|-----|------------|------------|----------------|------------|---------------|------------|------------|------------|------------|---------------|--------------------|
> | Llama3 14B | —   | 0.7987     | 0.7886     | 0.6788         | 0.8283     | 0.6024        | 0.965      | 0.4325     | 0.7720     | 0.7333     | 10.7807       | 3.6684             |
> | SVD-LLM   | 0.2 | 0.7622     | 0.6762     | 0.6914         | 0.6978     | 0.4684        | 0.910      | 0.4287     | 0.6249     | 0.6575     | 17.9817       | 4.1367             |
> | CoSpaDi| 0.2 | **0.7726** | **0.7172** | **0.7204**     | **0.7327** | **0.5128**    | **0.919**  | **0.4306** | **0.6580** | **0.6829** | **16.0535**   | **3.3936**         |
> | SVDLLM   | 0.4 | 0.6741     | 0.4869     | 0.4603         | 0.4848     | 0.3072        | 0.806      | 0.3665     | 0.3683     | 0.4943     | 35.7818       | 18.1019            |
> | CoSpaDi| 0.4 | **0.6953** | **0.5229** | **0.5731**     | **0.5324** | **0.3302**    | **0.844**  | **0.3751** | **0.4362** | **0.5387** | **32.6256**   | **8.7291**         |
>
> We will add these experiments in the revised manuscript, along with the corresponding analysis.

---

> ### Author Response · Authors · 2025-11-22
> **Part 2. Addressing weaknesses and questions**
>
> **4. Comparison with Dobi-SVD.**
>
> We appreciate the suggestion to include Dobi-SVD, but we believe it is not a directly comparable baseline for several reasons:
>
> * **Dynamic vs. Static compression ratio:** Dobi-SVD dynamically allocates per-layer rank budgets which addresses the problem that some layers are more crucial for the model performance, whereas CoSpaDi keeps the compression ratio fixed. We keep this direction as a further extension of our work.
> * **Training-free vs. training-required:** Dobi-SVD relies on full-network backpropagation to optimize layer-wise compression ratios with respect to the model’s target loss. In contrast, CoSpaDi is entirely training-free: it operates on a per-matrix basis using only a small calibration set, without joint optimization across layers or reference to the downstream loss.
>
> * **Weight Overparameterization:** The model compression achieved by Dobi-SVD is not attributed to the low-rank approximation of the model's weights. In fact, this method relies on an overparametrized representation of such weights. Inspection of the official implementation shows that matrices such as $4096 \times 4096$ are decomposed into $4096 \times 3266$ and $3266 \times 4096$, increasing parameters from 16.8M to 26.8M. The effective compression ratio for the 7B model at claimed 0.2 compression is approximately $-0.60 \pm 0.03$, indicating overparameterization before encoding. The compression is achieved with a quantization (so-called remapping), which places it closer to the quantization-based rather low-rank matrix approximation techniques.
> * **Compression mechanism:** Since Dobi-SVD requires the dequantization, the main benefit is achieved on the storage of the model rather than its deployment stage. Current authors' implementation requires custom kernels for acceleration which were not provided, thus performing full decompression prior to inference. In contrast CoSpaDi relies on widely supported GEMM which doesn't add any decoding overhead. Therefore, CoSpaDi provides a more practical approach for deployment-focused model compression, whereas Dobi-SVD excels primarily in storage efficiency.
>
> * **Complementarity:** Techniques for dynamic rank allocation and multi-layer budgeting, as well as advanced quantization strategies of the compressed weights are complementary to our framework and could be integrated with CoSpaDi, which we plan to investigate in our future work.
>
> * **Reproducibility considerations:** For completeness, we also attempted to reproduce the evaluation results of Dobi-SVD using the publicly available checkpoints for Llama-7B at compression ratios [0.2](https://huggingface.co/Qinsi1/DobiSVD-Llama-7b-hf-0.8), [0.4](https://huggingface.co/Qinsi1/DobiSVD-Llama-7b-hf-0.6) and [0.6](https://huggingface.co/Qinsi1/DobiSVD-Llama-7b-hf-0.4) with remapping and found several discrepancies with the reported results. Strictly following the setup steps, the official evaluation script produced numerical overflow for perplexity estimation on WikiText2, so we adopted the evaluation pipeline of SVD-LLM, which provided us with the reported WikiText2 perplexity for 0.2 compression. Using the authors' recommended environment configuration, we nevertheless observed noticeable differences between the reproduced accuracies and the results reported in the paper. We do not interpret this as an error in the method; rather, it indicates that the publicly released artifacts do not currently allow us to reproduce the original reported performance.
>
> **Table R3.3.** Reproduction results for Dobi-SVD
>
> | Method            | CR  | WikiText | PTB    | C4     | Openb. | Arc_e | WinoG. | HellaS. | Arc_c | PIQA  | MathQA | Average |
> |-------------------|-----|----------|--------|--------|--------|-------|--------|---------|-------|-------|--------|---------|
> | Dobi-SVD (paper)  | 0.2 | 6.08     | 15.39  | 7.83   | 0.27   | 0.65  | 0.68   | 0.54    | 0.37  | 0.77  | 0.27   | 0.50    |
> | Dobi-SVD  (repr.) | 0.2 | 6.08     | 32.79  | 8.49   | 0.33   | 0.74  | 0.69   | 0.55    | 0.40  | 0.77  | 0.26   | 0.53    |
> | Dobi-SVD (paper)  | 0.4 | 8.12     | 43.85  | 12.63  | 0.28   | 0.65  | 0.62   | 0.45    | 0.32  | 0.72  | 0.25   | 0.47    |
> | Dobi-SVD  (repr.) | 0.4 | 7.68     | 53.23  | 12.84  | 0.28   | 0.66  | 0.67   | 0.48    | 0.34  | 0.72  | 0.25   | 0.49    |
> | Dobi-SVD (paper)  | 0.6 | 9.95     | 67.62  | 17.94  | 0.23   | 0.52  | 0.56   | 0.38    | 0.24  | 0.65  | 0.23   | 0.40    |
> | Dobi-SVD (repr.)  | 0.6 | 14.98    | 205.25 | 46.16  | 0.20   | 0.41  | 0.58   | 0.34    | 0.23  | 0.60  | 0.22   | 0.37    |
>
> In summary we argue that as Dobi-SVD is mostly relying on quantization and efficient data formats rather low-rank matrix approximation **the direct comparison with our work is not applicable**.

---

> ### Author Response · Authors · 2025-11-22
> **Part 3. Addressing weaknesses and questions**
>
> **5. Unstructured pruning methods.**
>
> We thank the reviewer for suggesting the comparison with pruning-based methods (SparseGPT, Wanda). These methods **target inference speedups via sparsity**, not LLM weight compression, and therefore are not directly comparable. Importantly, unstructured pruning does not reduce the number of parameters stored: pruned models must still store both surviving values and their indices. For typical transformer dimensions, this results in:
> $$
> \text{per-weight cost} = 16 \text{ bits (value)} + [16{-}32] \text{ bits (index)} = [32{-}48] \text{ bits per retained weight}.
> $$
> At 50 % sparsity, total storage is $[16{-}24]$ bits per original weight - often **comparable to or larger** than dense bf16 storage. Furthermore, unstructured sparsity **requires specialized hardware support**, whereas CoSpaDi's structured sparse-dense operations are compatible with widely available GEMM kernels.
>
> **5. Rationale for choosing K-SVD**
> Our primary goal in this work is to demonstrate that dictionary learning with sparse coding can serve as a powerful alternative to conventional low-rank approximations for weight compression. To this end, we adopted K-SVD as a representative and widely used dictionary learning algorithm. As highlighted in the main paper (L218-219), K-SVD serves as a classical and well-established method within the dictionary learning literature, deliberately selected for its algorithmic transparency and reproducibility rather than its position as the SoTA solution; indeed, a variety of more advanced or specialized dictionary learning techniques exist.
>
> We view the exploration of alternative algorithms such as online, stochastic, or task-adaptive dictionary learning as an important and promising direction for future work. However, conducting a systematic comparison among these variants would substantially expand the scope of the study and detract from our central contribution: establishing the viability and advantages of the dictionary learning paradigm itself.
>
> **6. Real world questions answering**
>
> Below we provide example on Qwen3-8B compressed with SVD-LLM and CoSpaDi with 0.2 compression:
>
> **Question:**
> ```
> Mary does her grocery shopping on Saturday. She does her shopping only at a specific store where she is allowed a credit of $100, which must be paid in full before her next shopping trip. That week she spent the full credit limit and paid $15 of it on Tuesday and $23 of it on Thursday. How much credit will Mary need to pay before her next shopping trip?
> ```
>
> **Uncompressed model**:
> ```
> Let's break down the problem step by step:
>
> - **Credit Limit:** $100
> - **Amount spent:** The full credit limit — so **$100** was used.
> - **Payments made:**
>   - **Tuesday:** $15
>   - **Thursday:** $23
>
> ### Step 1: Total amount paid so far:
> $$
> 15 + 23 = 38 \text{ dollars}
> $$
>
> ### Step 2: Remaining balance:
> $$
> 100 - 38 = 62 \text{ dollars}
> $$
>
> ### Final Answer:
> Mary still needs to pay **$62** before her next shopping trip.
>
> ✅ **Answer: $62**
> ```
>
> **SVD-LLM answer:**
> ```
> Okay, let me try to figure this out for Mary.
>
> So Mary goes to her grocery store every Saturday. She's got a credit limit of $100 that she can use each time she shops. The thing is, she has to pay off the entire credit before she comes back again. So if she uses up all $100 worth of credit during one shopping trip, she needs to have that $100 paid by the next time she shops.
> Now, Mary went to the store last Saturday, and used up all $100 of credit. She paid $15 of it on Tuesday and $23 of it on Thursday. Wait, so how much did she still owe?
> Well, I guess the total amount she had to pay was $100. She paid $15 on Tuesday, then $23 on Thursday. Let me add those two amounts together: $15 plus $23 equals $38. So $38 was paid, but $62 remains unpaid.
> ...
> I think that’s pretty straightforward. If Mary has any questions about her account, I’d suggest she contact the store directly for clarification. But overall, I feel confident that the answer here is $62.
> ```
>
> **CoSpaDi answer:**
> ```
> Okay, let me try to figure this out.
>
> So Mary has a $100 credit limit that she can use for her groceries. She uses up all of it during the week, right? But she's not paying it off completely yet because she only paid $15 on Tuesday and $23 on Thursday. Wait, so how much more do they owe?
>
> Let me think. If she used up $100 total, then she needs to pay back $100. So far she’s paid $15 + $23 = $38. Let me add those two numbers together.
> $15 plus $23 equals... $38. So $100 minus $38 means she owes $62 more. So she needs to pay $62 before she can go shopping again.
> ...
> I don’t see anything else mentioned here. So $62 is what she needs to pay before her next shopping trip.
> So, the answer would be $62.
> ```
>
> Overall, we observe that models compressed at this ratio perform comparably to the original, whereas further compression leads to unstable behavior, including output cycling and hallucination, significantly undermining their practical applicability.

---

> ### Author Response · Authors · 2025-11-22
> **Part 4. Addressing weaknesses and questions**
>
> **Poor contribution of the paper**
>
> We respectfully disagree with the reviewer’s undervaluation of our contribution. While dictionary learning and sparse coding are classical tools in signal processing, their principled integration into training-free, activation-aware, structured compression of LLM weights with provable advantages over low-rank baselines is entirely novel and challenges entrenched assumptions in the field. Our core contributions are:
>
> * Reframing low-rank LLM compression as a rigid shared-subspace model, and showing analytically (Sec. 3.1) that its optimal solution is equivalent to PCA over weight columns - a perspective rarely emphasized, and one that exposes its expressivity limitations.
> Introducing union-of-subspaces modeling via sparse dictionary learning (SDL) to LLM weight compression - not merely as a plug-in replacement, but as a structured sparse factorization (dense dictionary $\times$ column-sparse codes), enabling each weight column to activate its own subspace. This relaxes orthogonality constraints and overcomes the “single-basis bottleneck” of SVD-based methods.
>
> * Proposing the first calibration-guided, training-free SDL framework for LLMs (CoSpaDi):
>   - We extend data-aware low-rank ideas (e.g., SVD-LLM) to *dictionary learning*, minimizing functional reconstruction error (via activation matching), not just weight approximation (Eq. 6-8).
>   - We support *both* *per-layer* and *cross-layer* sharing (inspired by Basis Sharing, but now with sparsity), enabling flexible deployment trade-offs (memory vs. fidelity).
>
> * Empirically establishing CoSpaDi as a new SOTA among matrix factorization methods, consistently outperforming modern baselines (SVD-LLM, ASVD, Basis Sharing, ReplaceMe, LLM-Pruner) across model families (LLaMA-3, Qwen), sizes (0.6B-8B), and compression regimes (20%-50%) without fine-tuning (Tables 2-5).
>
> * Crucially, no prior work has demonstrated that moving from global low-rank to local sparse subspaces - via training-free, activation-matched dictionary learning - yields such consistent, significant gains in LLM compression. This challenges the community’s long-standing reliance on SVD as the default factorization. We believe CoSpaDi opens a new design space for efficient, structured, and hardware-cooperative LLM deployment.
>
> **Final remarks**
> We hope our responses have addressed your concerns in full. If there are any remaining points that would benefit from clarification, we would greatly appreciate your additional feedback so we can ensure the revision fully meets your expectations.

---

> ### Author Response · Authors · 2025-11-26
> **Response to Reviewer mAYc**
>
> We would like to thank you again for your detailed review. We have responded to all points raised in your report and would be grateful for any follow-up comments or questions, in case there are any remaining concerns or questions from your side.

---

### Official Review · Reviewer_XpBJ · 2025-11-01

**Soundness:** 3
**Presentation:** 4
**Contribution:** 3
**Rating:** 6
**Confidence:** 3

**Summary:**

The paper proposes an LLM compression method that replaces low-rank approximation with sparse factorization (each weight matrix is represented with a dense dictionary and a column-sparse coefficient matrix) optimized with alternating steps of K-SVD. The authors show that this representation is more expressive and leads to a smaller approximation error.

**Strengths:**

1. **Novelty.** The authors apply dictionary learning, a technique known from signal and image processing, to improve on top of standard low-rank approximation while compressing LLM weights. They evaluate their method on a variety of models and benchmarks.
2. **Significance.** The paper addresses the problem of reducing the size of LLM parameters, which is crucial for deploying these models.
3. **Clarity.** The paper is well-written and describes the proposed method in a clear way.

**Weaknesses:**

1. **Compression time.** While L468 mentions that the current implementation is not optimal, it would be helpful to know how the compression time scales with model dimensions (now and with a potentially faster implementation), as well as how long it takes to compress the models used for the experiments.
3. **Only small models are evaluated.** The paper only mentions experiments on 1-8B models. If possible, it might be useful to evaluate larger models to make sure that the method still outperforms the baselines at scale.
2. **Optimal $\rho = k/s$ ratio.** Figure 2 doesn't clearly show that $\rho = 2$ is optimal (instead, $\rho = 1.6$ or $1.8$ seem the best), though I acknowledge that choosing a better $\rho$ would only make the proposed method stronger. Still, this figure is generally hard to read, and it might be useful to split it into three figures (one for each compression ratio) to highlight the accuracy/perplexity differences.
4. **Typos.** The paper often uses \cite{} in place of \citep{}.

**Questions:**

1. How does the compression time scale with model dimensions? How long does it take to compress the models used for the experiments?

---

> ### Author Response · Authors · 2025-11-22
> **Part 1. Addressing weaknesses and questions**
>
> We sincerely thank the reviewer for their thoughtful and encouraging assessment of our work. We are especially grateful that they recognized the novelty of our approach, the significance of the problem we address, and the clarity of our presentation. We also greatly appreciate their constructive suggestions, which have provided valuable guidance for improving the paper. Below, we address each point in detail and outline the revisions we will make in response.
>
> **1. Compression time.**
>
> We thank the reviewer for pointing out the absence of compression-time reporting. Table R2 summarizes the wall-clock time required to compress Qwen3 models across several parameter scales and compression ratios, using 8$\times$A100 GPUs for all runs.
>
> **Table R2.1.** Compression time (hh:mm) for Qwen3 models across compression ratios, using 8$\times$A100 GPUs.
> | Model       | CR = 0.2 | CR = 0.3 | CR = 0.4 | CR = 0.5 |
> |-------------|----------|----------|----------|----------|
> | Qwen3 0.6B  | 40m      | 35m      | 30m      | 26m      |
> | Qwen3 4B    | 6h 12m   | 5h 16m   | 4h 24m   | 3h 35m   |
> | Qwen3 8B    | 16h 33m  | 14h 04m  | 11h 33m  | 9h 54m   |
>
> These measurements indicate that compression time grows approximately linearly with the total number of parameters processed. As noted in the Limitations section of the manuscript, our current implementation employs sequential K-SVD updates and OMP-based sparse coding, both of which can be substantially accelerated. We identify several promising avenues for improving efficiency in future versions of CoSpaDi, including:
>
> * **Mini-batch and online dictionary learning.** Stochastic and online variants of dictionary learning, which update the dictionary using one sample or a small mini-batch at a time, have been shown to scale gracefully to very large datasets [A, B].
> * **Parallel and GPU-based K-SVD.** Parallel K-SVD formulations and GPU implementations can significantly accelerate both the sparse coding and dictionary update steps [C].
> * **Scalable alternating-optimization variants of K-SVD.** Recent work such as Double-Batch K-SVD (DB-KSVD) proposes scalable alternating-optimization schemes tailored to high-dimensional embedding spaces, explicitly targeting settings similar to modern Transformer models [D].
> * **Randomized algorithms for large-scale dictionary learning.** Randomized techniques for dictionary learning exploit approximate low-rank structure to reduce computational cost while maintaining high-quality dictionaries, and provide another promising direction for accelerating our framework [E].
>
> We would like to emphasize that the primary objective of this work is to demonstrate the effectiveness of sparse dictionary learning for LLM compression as a stronger alternative to low-rank approximation strategies, rather than to fully optimize all aspects of the dictionary learning pipeline itself. Nevertheless, we view efficiency-oriented improvements along the lines above as valuable and complementary future work on CoSpaDi.
>
> [A] - Mairal, Julien, et al. "Online dictionary learning for sparse coding." Proceedings of the 26th annual international conference on machine learning. 2009.
>
> [B] - Mensch, Arthur, et al. "Dictionary learning for massive matrix factorization." International Conference on Machine Learning. PMLR, 2016.
>
> [C] - He, Lu, et al. "Scalable 2D K-SVD parallel algorithm for dictionary learning on GPUs." Proceedings of the ACM International Conference on Computing Frontiers. 2016.
>
> [D] - Valentin, Romeo, et al. "DB-KSVD: Scalable Alternating Optimization for Disentangling High-Dimensional Embedding Spaces." arXiv preprint arXiv:2505.18441 (2025).
>
> [E] - Wu, Gang, and Jiali Yang. "Randomized algorithms for large-scale dictionary learning." Neural Networks 179 (2024): 106628.

---

> ### Author Response · Authors · 2025-11-22
> **Part 2. Addressing weaknesses and questions**
>
> **2. Larger models.**
>
> We fully agree that validation on larger models is crucial for demonstrating the scalability and practical relevance of our approach. At the same time, we would like to contextualize our experimental scope within the broader LLM-compression literature. Recent training-free or minimally-trained compression methods - such as ASVD (evaluated primarily on 7-13B models), SVD-LLM (3-7B), Dobi-SVD (7-13B), and LLM-Pruner (6-13B) - have predominantly focused on models in the 7B-13B range. This is largely because models at these scales already exhibit substantial internal redundancy, while remaining computationally feasible to compress and evaluate across many variants.
>
> To address the reviewer’s concern within limited time and computational budget, we have conducted new *preliminary* experiments on **Qwen3-14B** at compression ratios of 0.2,  and 0.4. These results consistently show that CoSpaDi continues to outperform SVD-LLM at this larger scale, supporting the method’s applicability beyond the 1B-8B range explored in the original submission.
>
> | Method   | CR  | PIQA       | HellaSwag  | Lambada OpenAI | ARC-easy   | ARC-challenge | SciQ       | Race       | MMLU       | Avg. Acc.  | Wikitext (PPL) | Lambada (PPL) |
> |----------|-----|------------|------------|----------------|------------|---------------|------------|------------|------------|------------|---------------|--------------------|
> | Llama3 14B | —   | 0.7987     | 0.7886     | 0.6788         | 0.8283     | 0.6024        | 0.965      | 0.4325     | 0.7720     | 0.7333     | 10.7807       | 3.6684             |
> | SVD-LLM   | 0.2 | 0.7622     | 0.6762     | 0.6914         | 0.6978     | 0.4684        | 0.910      | 0.4287     | 0.6249     | 0.6575     | 17.9817       | 4.1367             |
> | CoSpaDi| 0.2 | **0.7726** | **0.7172** | **0.7204**     | **0.7327** | **0.5128**    | **0.919**  | **0.4306** | **0.6580** | **0.6829** | **16.0535**   | **3.3936**         |
> | SVDLLM   | 0.4 | 0.6741     | 0.4869     | 0.4603         | 0.4848     | 0.3072        | 0.806      | 0.3665     | 0.3683     | 0.4943     | 35.7818       | 18.1019            |
> | CoSpaDi| 0.4 | **0.6953** | **0.5229** | **0.5731**     | **0.5324** | **0.3302**    | **0.844**  | **0.3751** | **0.4362** | **0.5387** | **32.6256**   | **8.7291**         |
>
> We will add these experiments across multiple compression ratios for Qwen3-14B in the revised manuscript, along with the corresponding analysis.
>
> **3. Clarification on the optimal $k/s$ ratio.**
>
> We appreciate the reviewer’s observation. The original Figure 2 did not clearly illustrate that the region around $k/s=2$ yields near-optimal performance across all tested compression ratios. We selected this value according to our preliminary investigation on 0.2 compression for Llama 3.2 1B and fixed it for all our experiments. We agree that the per-model optimization or even dynamic per-layer estimation of the k/s ratio will further strengthen the CoSpaDi performance. This direction lies beyond the scope of the present work but represents a promising avenue for future research.
>
> **4. Citation formatting.**
> Thank you for noting inconsistencies in citation style. All references were corrected to use the proper citation command in the  **uploaded revised version of the manuscript**.
>
> **Closing remarks**
>
> We believe that our answers resolve the issues you have highlighted, and we are glad to provide further clarification if needed.

---

> ### Author Response · Authors · 2025-11-26
> **Response to Reviewer XpBJ**
>
> Thank you for your thoughtful review. We have addressed all items in our rebuttal and would be happy to clarify any points or provide additional information if helpful.

---

### Official Review · Reviewer_DoQo · 2025-11-01

**Soundness:** 3
**Presentation:** 2
**Contribution:** 3
**Rating:** 6
**Confidence:** 4

**Summary:**

This paper proposes CoSpaDi, a training-free compression framework for large language models that employs sparse dictionary learning as an alternative to conventional low-rank decomposition methods. The key innovation lies in representing each weight matrix as a product of a dense dictionary and a column-sparse coefficient matrix, enabling a union-of-subspaces representation where different weight columns can be approximated in distinct subspaces. The method incorporates calibration-guided optimization to minimize functional reconstruction error in the activation space rather than direct weight approximation. The authors provide comprehensive theoretical derivations and demonstrate superior performance over state-of-the-art low-rank baselines (SVD-LLM, ASVD, Basis Sharing) on Llama and Qwen models at 20-50% compression ratios, evaluated on standard language understanding benchmarks. The framework supports both per-layer and cross-layer compression scenarios and is compatible with post-training quantization.

**Strengths:**

1. **Strong theoretical foundation:** The paper provides rigorous mathematical derivations connecting low-rank approximation to PCA (Appendix A.1), deriving the data-aware optimization objective (Appendix A.2), and establishing the compression ratio formulation (Eq. 9-10). The theoretical treatment is comprehensive and well-presented, clearly distinguishing the proposed union-of-subspaces representation from traditional single-subspace low-rank methods.

2. **State-of-the-art model selection:** The experimental evaluation employs very recent and relevant models including Llama-3.2, Llama-3, Qwen-3, and Llama-2 across different scales (0.6B to 8B parameters), demonstrating the method's applicability to modern LLM architectures.

3. **Consistent performance gains:** CoSpaDi demonstrates substantial improvements over SVD-based baselines across multiple compression ratios, with particularly strong results at moderate compression levels (20-30%), showing both better accuracy and lower perplexity.

**Weaknesses:**

1. **Limited evaluation benchmarks:** The evaluation relies heavily on relatively outdated tasks (PIQA, HellaSwag, ARC, SciQ from 2017-2019, MMLU from 2021). For a paper targeting state-of-the-art models like Llama-3 and Qwen-3, the benchmark suite should include more recent and practically relevant evaluations:

2. **Poor presentation quality and readability:**
- Figure 1 is particularly problematic: The use of gradient colors is visually unappealing and does not effectively convey the conceptual differences between low-rank decomposition and dictionary learning. The color scheme is distracting rather than informative.
- Inconsistent figure quality: Figures 2-7 vary significantly in style and quality, lacking a unified visual language
- Table formatting: Tables 2-6 are dense and difficult to parse quickly; better use of spacing, grouping, and highlighting would improve readability
- Complex notation overload: The main text introduces substantial notation (W, D, S, L, X, Y, etc.) without sufficient visual aids or intuitive explanations for readers less familiar with dictionary learning

**Questions:**

1. **Benchmark selection rationale:** [Key Concerns] Can the authors explain why more recent and practical benchmarks were not included? For instance, technical reports for Llama-3, 3.2 and Qwen-3 emphasize code generation, mathematical reasoning, and domain-specific capabilities. Would including these tasks change the relative performance of CoSpaDi vs. baselines?

2. **Dictionary interpretability:** Do the learned dictionary atoms have any semantic interpretation? For example, do certain atoms consistently activate for specific linguistic phenomena or token types?

3. **Coefficient quantization:** Table 1 shows mantissa truncation results. Have the authors explored other quantization schemes (e.g., k-means quantization, learned quantization) that might preserve more information in the sparse coefficients?

---

> ### Author Response · Authors · 2025-11-22
> **Part 1. Addressing weaknesses and questions**
>
> We sincerely thank the reviewer for the constructive and detailed feedback. We appreciate the positive assessment of our theoretical formulation, motivation, and demonstrated practical improvements over low-rank methods. Below we provide point-by-point responses to the raised concerns.
>
> **1. Benchmark selection rationale.**
>
> We thank the reviewer for suggesting the inclusion of more modern benchmarks capturing practical reasoning, coding, and instruction-following capabilities. Our initial benchmark suite (PIQA, HellaSwag, ARC, SciQ, Race, MMLU, WikiText, LAMBADA) follows established conventions in training-free compression work (e.g., SVD-LLM, ASVD, pruning-based baselines), enabling a direct and fair comparison under standard evaluation protocols.
>
> Nevertheless, following the reviewer’s recommendation, we expanded our evaluation with a complementary set of widely adopted modern tasks aligned with capabilities emphasized in the Llama-3 and Qwen-3 technical reports, while remaining computationally feasible across many compressed variants. Specifically:
> (i) for mathematical reasoning we added **MathQA** (2019), **MATH** (2021) and **GSM8K** (2021),
> (ii) for code generation we added **HumanEval** (2021), and
> (iii) for instruction following we included the **Ifeval** (2023) dataset.
>
> To strengthen the evaluation, we incorporated additional contemporary benchmarks: **BBH** (2022), **GPQA** (2023), **MUSR** (2023), and **MMLU-Pro** (2024), which further demonstrate that CoSpaDi consistently outperforms strong low-rank-based baseline.
>
> **Table R1.** Performance comparison of CoSpaDi vs. SVD-LLM on Qwen3-8B at different compression ratios on modern benchmarks. Best results are in **bold**.
>
> | Method        | CR  | MathQA | GSM8K  | IfeVal | BBH   | MATH  | GPQA  | MUSR  | MMLU-Pro |
> |---------------|-----|--------|--------|--------|-------|-------|-------|-------|----------|
> | **Qwen3 8B**  | —   | 49.31  | 87.64  | 39.21  | 60.86 | 52.57 | 36.16 | 43.12 | 47.72    |
> |               |     |        |        |        |       |       |       |       |          |
> | SVD-LLM       | 0.2 | 30.92  | 32.37  | 25.54  | 41.00 | 1.06  | 28.36 | 39.81 | 26.30    |
> | **CoSpaDi**   | 0.2 | **34.17** | **50.04** | **28.90** | **45.25** | **1.96** | **28.61** | **42.06** | **31.46** |
> |               |     |        |        |        |       |       |       |       |          |
> | SVD-LLM       | 0.3 | 27.74  | 3.26   | 22.90  | 34.42 | 0.98  | **25.59** | **41.40** | 18.82    |
> | **CoSpaDi**   | 0.3 | **30.99** | **13.19** | **25.18** | **38.22** | 0.98  | 24.75 | 38.36 | **22.81** |
> |               |     |        |        |        |       |       |       |       |          |
> | SVD-LLM       | 0.4 | 23.55  | 0.00   | 22.66  | 30.24 | **0.83** | 23.07 | 37.70 | 11.55    |
> | **CoSpaDi**   | 0.4 | **24.79** | **0.53** | **26.14** | **32.72** | 0.76  | **26.01** | **38.10** | **16.74** |
> |               |     |        |        |        |       |       |       |       |          |
> | SVD-LLM       | 0.5 | **22.35** | 0.00   | 22.06  | 29.06 | 0.38  | **26.26** | **36.64** | 11.10    |
> | **CoSpaDi**   | 0.5 | 22.11  | 0.00   | **24.34** | **29.94** | **0.91** | 26.17 | **36.64** | **11.26** |
>
> These additional results consistently show that CoSpaDi in most cases achieves substantial improvements over SVD-LLM even on modern and challenging benchmarks, supporting the method’s robustness and generalization ability.
>
> **2. Manuscript readability and presentation.**
>
> We appreciate the reviewer’s suggestions regarding figure aesthetics, table readability, and notation clarity. In the **uploaded revised manuscript**:
> * Figure 1 has been fully redesigned with a lighter, cleaner color palette and improved labeling, ensuring consistency with notation used throughout the method description.
> * All tables have been reformatted with improved spacing, grouping, and readability.
>
> We welcome any additional suggestions for improving figure style and visual consistency.
>
> **3. Dictionary interpretability.**
>
> We agree that interpreting dictionary atoms is an intriguing direction. Preliminary inspection suggests nontrivial activation patterns, but a rigorous interpretability analysis (e.g., linguistic specialization, structural roles, alignment with MoE-like behaviors) is beyond the scope of this compression-focused work. We plan to explore these phenomena in future research.

---

> ### Author Response · Authors · 2025-11-22
> **Part 2. Addressing weaknesses and questions**
>
> **4. Coefficient quantization.**
>
> We appreciate the reviewer’s comments on coefficient quantization. In the submitted paper, we used simple bfloat16 mantissa truncation as a proof-of-concept to demonstrate compatibility between sparse dictionary learning and quantization. As shown in Table 1 (main paper), reducing mantissa precision from 16 to 14 bits introduces negligible performance degradation across all evaluated compression ratios, while eliminating index-storage overhead.
>
> We agree that more advanced quantization strategies, e.g., $k$-means quantization, learned quantizers, or mixed-precision assignments could yield even better trade-offs. Preliminary explorations suggest that uniform quantization below 14 bits harms performance, indicating that more adaptive approaches are promising future directions. We plan to incorporate such techniques in follow-up work to further boost CoSpaDi's compression efficiency.
>
> **Closing remarks**
>
> We hope that our responses adequately address all of your concerns, and we welcome any further questions or clarifications that may support your assessment.

---

> > ### Comment · Reviewer_DoQo · 2025-11-26
> > **All my question have been solved. I decide to keep the rating.**
> >
> > All my question have been solved. I decide to keep the rating. I am particularly looking forward to the public release of your reproducible GitHub repository,

---

> > > ### Author Response · Authors · 2025-11-26
> > > **Response to Reviewer DoQo**
> > >
> > > Dear reviewer DoQo,
> > >
> > > Thank you very much for the thoughtful follow-up and for taking the time to review our responses.
> > >
> > > We are glad to hear that all your questions have been resolved, and we appreciate your interest in the forthcoming public release of our fully reproducible GitHub repository.
> > >
> > > If you feel that the revisions and additional experiments sufficiently address the concerns raised in your review, we would be grateful if you would consider updating your rating accordingly.

---

### Meta-Review · Area_Chair_R7o8 · 2025-12-21

**Summary:**

This submission proposes CoSpaDi, a training-free LLM compression approach based on sparse dictionary learning, positioned as a more expressive alternative to standard low-rank factorization, and evaluated primarily against SVD-style baselines. Reviews raise concerns around practical deployment and evaluation completeness, including end-to-end inference latency, substantial performance drop at aggressive compression, insufficient baselines, which remain valid after rebuttal.

**Reviewer Concerns:**

Concerns on limited evaluation benchmarks, compression time, scalability, presentation, are resolved during rebuttal.  Concerns on end-to-end inference latency, large performance drop at aggressive compression, insufficient baselines remain valid.

**Reviewer Scores:**

The two reviewers that gave 6 most likely to remain borderline. The reviewer who gave 2 most likely remain negative towards the paper as the concerns are not fully addressed.

---

### Decision · Program_Chairs · 2026-01-26

Reject